



**Title:** Development of a harmonized soil profile analytical database for Europe: A
resource for supporting regional soil management
**Authors:**
Jeppe Aagaard Kristensen[1,2*‡], Thomas Balstrøm[2], Robert J.A. Jones[3], Arwyn Jones[4], Luca
Montanarella[4], Panos Panagos[4], and Henrik Breuning-Madsen[2†‡].
**Affiliations:**
[1]Department of Physical Geography and Ecosystem Science, Lund University, Sölvegatan 12,
223 62 Lund, Sweden.
[2]Department of Geosciences and Natural Resource Management, University of Copenhagen,
1350 Copenhagen K, Denmark.
[3]School of Energy, Environment and AgriFood, Cranfield University, College Road, Cranfield,
MK43 0AL, UK.
[4]European Commission, DG Joint Research Centre, Via E. Fermi 2749, 21027 Ispra (VA), Italy.
*Correspondence to jeppe.aa.kristensen@gmail.com
†Deceased
‡These authors contributed equally to this work.
**Running head:** A harmonized soil profile analytical database for Europe.




## Abstract

Soil mapping is an essential method to obtain a spatial overview of soil resources that are

increasingly threatened by environmental change and population pressure. Despite recent

advances in digital soil mapping techniques based on inference, such methods are still immature

for large-scale soil mapping. During the 1970s, 80s and 90s, soil scientists constructed a

harmonised soil map of Europe (1:1M) based on national soil maps. Despite this extraordinary

regional overview of the spatial distribution of European soil types, crude assumptions about soil

properties were necessary to translate the maps into thematic maps relevant for management. To

support modellers with analytical data connected to the soil map, the European Soil Bureau

commissioned the development of the Soil Profile Analytical Database for Europe (SPADE) in

the late 1980s. This database contains soil analytical data based on a standardised set of soil

analytical methods across the European countries. Here, we review the principles adopted for

developing the SPADE database during the past five decades, and the work towards fulfilling the

milestones of full geographic coverage for dominant soils in all the European countries (SPADE

level 1), and the addition of secondary soil types (SPADE level 2). We demonstrate the

application of the database by showing the distribution of the root zone capacity, and by

estimating the soil organic carbon (SOC) stocks to a depth of 1 m for EU-27 to $76 \times 10^{15}$ g. The

increased accuracy, potentially obtained by including secondary soil types (level 2), is

demonstrated in a case study of estimating SOC stocks in Denmark. In the lack of systematic

cross-European soil analysis schemes, integrating national soil maps and locally assessed

analytical data into a harmonised database is a powerful resource to support soil resources

management at regional and continental scales by providing a platform to guide sustainable soil

management and food production.






**Keywords**: EU soil map; SPADE; Soil data harmonisation; Soil organic carbon; Root zone
capacity

## Introduction

In a world subject to constant environmental change and increasing population pressure, soil
becomes an increasingly important but threatened resource (FAO 2015; Sustainable Food Trust
2015). This challenge must be met at multiple levels and scales; hence, accurate understanding of
the available resources at the appropriate scale is required (e.g. Robinson et al. 2017). In spite of
advances in digital soil mapping using remote sensing and geographical information systems to
infer soil properties (McBratney et al. 2003; Arrouays et al. 2014; Minasny and McBratney
2016; Zhang et al. 2017), we still lack adequate standardised methods for large scale soil
mapping. Furthermore, the existing methods are particularly challenged in densely vegetated
areas and for subsoil properties (Mulder et al. 2011), which are highly relevant for environmental
management and food production. A recent assessment of the implications of uncertainty in soil
data found that it could potentially offset climate change impact on future crop yields, due to the
dependence on soil type (Folberth et al. 2016).
During the last century, national soil maps were established in most European countries but they
were not harmonised across borders as they were based on specific national soil classification
systems (Morvan et al., 2008). Therefore, international soil classification systems were
developed during the 1960s and early 1970s to facilitate the construction of globally standardised



soil maps (FAO-Unesco 1974, SMSS/USDA/AID 1983). The FAO maps portrayed the soils
resources for each individual country as mapping units with a distinct set of soil types. The soil
types comprised three categories: dominant soils, associated soils, and inclusions. The dominant
soil type covered the largest proportion of the mapping unit; associated soils occupied 20% to
50% of the unit while the inclusions accounted for less than 20%. The maps were published with
an explanatory text describing the geology, geomorphology, land use and a map showing the
level of knowledge behind the map construction, i.e. the level of confidence (King et al. 1994).
In the beginning of the 1980s, the ten European Communities (EC) Member States elaborated
the FAO-Unesco approach to make an expanded and a more detailed version of the FAO-Unesco
(1974) system for the soil types present in their respective countries. Based on this, the EC
published seven soil maps (scale 1:1M, Commission of the European Communities, 1985). The
complete soil map of Europe was digitised by the end of the 1980s (Platou et al. 1989) as a part
of the EC financed CORINE programme (Briggs & Martin 1988), and quickly, it became an
important dataset in the forecasting of national crop yields across Europe by the European
Commission's Joint Research Centre's Monitoring Agriculture of Remote Sensing (MARS)
project (Vossen 1993). Subsequently, the EC soil map was used widely to underpin soil resource
assessments within the European Union (EU) including the mapping of carbon (C) stocks
(European Commission, 2005; Jones et al., 2005; Lugato et al., 2014), soil erosion risks (Kirkby
et al. 2008, Panagos et al. 2015), vulnerability to compaction (Jones et al. 2003, Schønning et al.
2015) and salinity (European Commission, 2005), as well as raising awareness and providing
education materials (e.g. European Commission, 2005).
Yet, such assessments are based on assumptions about each soil type's characteristics or
extrapolations from limited amounts of (often) country specific analytical data. Therefore,



incorporating national datasets into one uniform European database would dramatically increase
the quality of predictions and evaluations based on the EC Soil Map across Member State
borders. A global attempt to meet this challenge has led to the development of the Harmonized
World Soil Database (FAO/IIASA/ISRIC/ISSCAS/JRC 2012), but this database extracts its data
from Europe from the European Soil Database (v.2.0), which in turn is based on the soil profile
analytical database for Europe (SPADE). This paper demonstrates how this cornerstone in the
European Soil Data Centre (Panagos et al. 2012) was created based on soil physical and chemical
soil data provided by national stakeholders from each member state. Specifically, a database
containing estimated analytical data for all dominant soil types within the EU with full
geographical coverage (SPADE 14) was compiled. Furthermore, a level 2 database was
developed for a subset of countries, and a full coverage level 2 database (SPADE 18), will in the
years to come be expanded to cover the entire EU and surrounding countries. Finally, it is
demonstrated how the database can be used to obtain estimates of environmentally relevant soil
properties (e.g. root zone capacity and SOC-stocks).

**Establishing the Soil Profile Analytical Database of Europe framework (SPADE 1)**
A working group of Europe-wide soil specialists was formed to advise the Commission of the
European Communities on the establishment of a soil profile analytical database (SPADE)
connected to the EC soil map (Figure 1a). By the end of the 1980s, the Working Group proposed
that it should be based on four levels of analytical data (Breuning-Madsen 1989): Level 1 would
provide analytical data from a typical soil profile for the dominant soil typological unit (STU) in
each soil mapping unit (SMU), preferably on arable land; Level 2 would expand the database to
include a typical dataset for all STUs, including associated soils and inclusions; Level 3 would



be a further expansion to include soil analytical data for all soil types with a differentiation
between land uses; Level 4 would allow different soil analytical data for the same soil type
(STU) that occurs in different sub-regions (e.g. based on geology or geomorphology). (See
Figure 1b for a timeline).
Initially, two soil analytical databases were established; one containing estimated mean values
for typical soil profiles according to fixed soil analytical procedures provided by national
stakeholders (referred to as Proforma I), while another contains soil profile data measured using
established analytical procedures (referred to as Proforma II). The Proforma I database contains
data comparable across country borders while this is not always the case for the Proforma II
database. In order to make the database functional as soon as possible for the entire coverage
area, each Member State stakeholder was asked to deliver one full set of Proforma I (estimated)
analytical data for each dominant soil type (STU) in each of the soil mapping units (SMU)
delineated on the Soil Map of Europe (1:1M). Providing data for the Proforma II (measured)
database was optional. Where possible, the data should be provided for agricultural land, as the
primary aim of the database was to underpin large-scale assessments of agricultural land
management.
In 1993, Proforma I and II schemes (including guidelines) were sent to the stakeholders in order
to collect data for the individual countries; detailed guidelines for compilation of the SPADE 1
dataset was published by Breuning-Madsen and Jones (1995).
Subsequently, the SPADE 1 database was expanded to include data from the new EU Member
States but also from non-EU nations such as Albania, Norway and Switzerland. By the end of the
1990s, SPADE 1 was subject to a data quality assessment and scrutinised to identify missing data



and evaluate overall data reliability. Based on the recommendations presented at a European Soil
Bureau Network (ESBN) meeting in Vienna 1999, the national stakeholders were requested to
update their individual datasets. Meanwhile, only a few national stakeholders responded, which
left the SPADE 1 incomplete and not well suited for modelling at the European level.

**An attempt to populate SPADE with measured data (SPADE 2)**
Due to the limitations of SPADE-1, SPADE-2 was developed to derive appropriate soil profile
data to support, for example higher tier modelling of pesticide fate at the European level (Hollis
et al., 2006). Data were supplied from national data archives, similar to SPADE 1 Proforma II.
Despite the analytical methods differing between countries, the raw national data were
harmonised and validated to provide a single data file for use in conjunction with the existing
Soil Geographical Data Base of Europe (Platou et al. 1989). The primary soil properties required
for each soil were: Horizon nomenclature (e.g. A, E, B, C), upper and lower depth (cm), particle-
size distribution: clay, silt, total sand and content of at least 3 sand fractions, pH in water (1:2.5
soil:water), organic carbon content (%) and dry bulk density (g cm$^{-3}$).
The acquisition of data happened in two steps; first datasets were obtained from Belgium,
Luxembourg, Denmark, England and Wales, Finland, Germany, Italy, the Netherlands, Portugal
and Scotland (Hollis et al. 2006), and next  the database was expanded with data from Bulgaria,
Estonia, France, Hungary, Ireland, Romania, Slovakia, Spain, France and Ireland . The final
database (SPADE2v11) only exists as a beta version of collated datasets from the first and
second phases of soil profile data acquisition (Hannam et al. 2009). However, it was used to
estimate bulk densities for missing data in the later SPADE 14.




**Steps towards full geographical coverage (SPADE 8)**

In an effort to obtain a functional database with full spatial coverage for the EU, a small

specialist group from Denmark (Prof. Henrik Breuning-Madsen, Assoc.Prof. Thomas Balstrøm

and MSc. Mads Koue from the Institute of Geography, University of Copenhagen) undertook a

scrutiny of the national datasets in 2008 using error finding equations based on literature values,

expert judgements, and pedotransfer functions (Koue et al. 2008).

First, a quality check was conducted on all data. This process consisted of:

i)    cross-checking of interdependent variables (e.g. pH vs. base saturation or porosity vs.

        saturated water content); and

ii)   checking the plausibility of all values according to published theoretical or empirical

        values (e.g. for bulk density (BS) or C:N-values).

Examples of common questionable data were occurrences of bulk soil C:N values <5,

mismatches between BS and pH (e.g. BS>90% at pH<4.5), and volumetric water content at

saturation exceeding the porosity. Moreover, in several cases the sum of clay, silt and sand

fractions differed from 100 %. Based on this examination, implausible values were either

adjusted to plausible values or marked as unlikely based on predefined criteria. In terms of

spatial extent, it was only possible to link a soil analytical dataset for a dominant soil type to

approximately70% of the soil mapping units (SMU) in the area covered by the database, due to

missing data.



Following an ESBN meeting in Paris, December 2008, the reviewed SPADE 8 database was
discussed and the national evaluation reports, together with the country specific databases, were
sent to the national stakeholders with a request to i) review and change the existing data to
plausible values based on the expert scrutiny, and ii) estimate new datasets for the dominant soil
types without data based on their local expertise. The modifications received from the
stakeholders were incorporated in the SPADE 8 database that was renamed SPADE 11.
However, once again only few responses were received, which still left the database incomplete,
so SPADE 11 remained as unpublished work in progress.



**Establishing a SPADE for dominant soil types with full coverage of the EU (SPADE 14)**
Without further input from the national stakeholders, implausible data identified in SPADE 8
were estimated to make the Proforma I (level 1) database more functional for modelling. Thus, in
2014 and early 2015, the SPADE 8 database was updated by a working group consisting of the
current paper's authors.
Specifically, this work package had three key goals:

189       i: To implement the suggested improvements of the existing data in the SPADE database

190       suggested during the 2008 evaluation,



ii: To estimate values for the profiles lacking data (approximately 32% of the dominant
STUs) based on matching of similar soil types in neighbouring countries, the data in
SPADE 2, or other reference data sources.
iii: To update the existing SPADE database with the complete dataset after revision by
the national stakeholders.
The final SPADE14 database is publically available through JRC's European Soil Data Centre
(ESDAC) website (http://esdac.ec.europa.eu/).
Firstly, the questionable values identified in SPADE 8, but not corrected by stakeholders due to
passivity, were adjusted to fit theoretical or average values according to predefined equations or
guidelines (Breuning-Madsen et al. 2015). Secondly, data for profiles lacking stakeholder
estimated values were assigned by copying complete datasets from identical soil types in
neighbouring countries. If no matching profiles were identified, the search was extended to the
entire database. Thirdly, data for the remaining ~15% of the dominant soil types (STUs for
which no estimated data existed anywhere in the database) was created by adjusting existing data
from similar soil profiles, preferably from the country itself or neighbouring countries to
minimise confounding factors. The evaluation guidelines sent to the stakeholders during the
SPADE 14 evaluation (Breuning-Madsen et al. 2015) provided a detailed description of the
methodology. The entire database was quality controlled with the updated versions of equations
and guidelines used during the 2008 evaluation thus ensuring consistency across Member States.
Finally, the quality controlled national data where sent to each stakeholder for final checking and
revision before publication.
*Examples of correction guidelines*





For some parameters, no correction guidelines were specified during the 2008 evaluation, in
which case they weredeveloped during the 2014/15-evaluation. As an example, the estimation of
bulk density and volumetric water content are elaborated below.
*Bulk density*
Missing bulk density (BD) values were assigned the average of all measured values from the
SPADE 2 (Table 1). For soil horizons with organic matter (OM) content >10%, BD values were
calculated from the OM content grouped into 10% intervals. For soils with OM contents <5%,
BD values were averaged over depth intervals of 25 cm down to 100 cm. Deeper horizons were
assigned a value of 1.5 g cm$^{-3}$ unless geomorphology or overlying horizons indicated a
significantly different value. For soils with OM contents between 5 and 10%, the BD was
estimated a value in the range 1.1-1.2 g cm$^{-3}$ based on surrounding horizons and profiles.



*Volumetric water content (VWC)*
National stakeholders were requested to specify the water content at 1, 10, 100 and 1500 kPa
suction for each soil horizon enabling the calculation of functions such as root zone capacities. In
order to assign realistic data to missing estimates, we regressed (multivariate linear regression)
water retention data, i.e. VWC at 1, 10, 100 and 1500 kPa suction, from countries with complete
datasets against multiple explanatory variables; bulk density (BD), particle size fractions (TEXT,
<2 μm, 2-20 μm, 20-50 μm, 50-200 μm, 200-2000 μm) and organic matter content (OM, %).



Member States with complete estimated datasets were Belgium, United Kingdom (UK) and
Denmark. As data from DK were used for validation, the derived equations were based on data
from Belgium and the UK. Fluvisols were omitted as they often have complicated water
retention properties due to their geomorphological origin. Only 7 % (9 of 132) of the
observations from DK deviated more than 10% VWC from the 1:1 line between observed and
calculated values using the linear models. The adjusted correlation coefficients were 0.85, 0.86,
0.87 and 0.91 for $VWC_1$, $VWC_{10}$, $VWC_{100}$, and $VWC_{1000}$, respectively ($P < 0.001$), and the
resulting regression equations were:
$VWC_1 = (-27.653 \times BD + 1.463 \times OM + 0.208 \times TEXT_2 + 0.017 \times TEXT_{20} + 0.154 \times TEXT_{50} +$
$0.013 \times TEXT_{200} + 0.003 \times TEXT_{2000} + 57.783) \times BD$
$VWC_{10} = (-20.231 \times BD + 1.110 \times OM + 0.262 \times TEXT_2 + 0.029 \times TEXT_{20} + 0.193 \times TEXT_{50} -$
$0.026 \times TEXT_{200} - 0.072 \times TEXT_{2000} + 41.072) \times BD$
$VWC_{100} = (-4.246 \times BD + 1.356 \times OM + 0.335 \times TEXT_2 + 0.071 \times TEXT_{20} + 0.105 \times TEXT_{50} -$
$0.002 \times TEXT_{200} - 0.015 \times TEXT_{2000} + 8.380) \times BD$
$VWC_{1500} = (-0.330 \times BD + 1.088 \times OM + 0.358 \times TEXT_2 + 0.125 \times TEXT_{20} + 0.072 \times TEXT_{50} +$
$0.056 \times TEXT_{200} + 0.053 \times TEXT_{2000} - 4.719) \times BD$

*Traceability and quality check*
In order to ensure traceability of all proposed changes, we developed a colour coding system to
the Excel spreadsheets submitted to stakeholders that allowed them to identify what kind of
changes had been applied to each data element. Moreover, a tracing document keep track of



whether the dominating STUs contained original stakeholder estimated data, a dataset copied
from another profile in the database, or a dataset modified by the working group. For the latter, a
separate tracing document kept track of profiles and parameters modified to anticipate criticism
and corrections by national stakeholders. Finally, the data quality was evaluated as prior to the
modifications, and a new cross-database-check was introduced to make sure whether the topsoil
texture class specified in the estimated profile database matched the actual topsoil texture class
specified in the estimated horizon database. When inconsistencies were identified, the topsoil
texture class in the estimated horizon database was adjusted accordingly.

*Evaluating, updating and publishing the SPADE-14 database*
Table 2 provides an overview of the origin of the data for each country. The first column
(Original SPADE-8) shows how many profiles were available from both SPADE 1 and 8. The
second column (SPADE 14 - Profiles from other countries) shows how many profiles were
copied from other countries, and the third column (SPADE 14 - Modified profiles) shows how
many profiles that were created by the working group by adjusting existing profiles in order to
complete the national datasets.



Overall, the SPADE 18 (level 2) database contains soil analytical data from 1831 profiles which
is about 60% more than the number of profiles in SPADE 14 (level 1) containing soil analytical



data from 1078 profiles, which is, almost a doubling of the number of profiles available in
SPADE 1 and 8. Most of the profiles originally lacking data had allocated datasets from
complete profiles from other countries. Yet, ~15% of the dominant profiles specified by soil type
and texture were not present in either SPADE 1 nor 8 and had to be constructed by modifying
other existing profile datasets to fit the required soil classification. Eight countries did not deliver
data to SPADE 1 nor 8. Thus, datasets for these countries were exclusively based on imported or
constructed datasets. Stakeholders have been notified throughout this project that they may
update their national datasets at any time by contacting the responsible ESDAC office.

**Creating a pilot version of the SPADE 18 level 2 database (SPADE-18)**
As described previously, the SPADE framework has four levels. The level 2 database contains
the same type of analytical data as the level 1 database, but in addition to the dominating soil
types, the inclusions and associations have been assigned estimated analytical data. This allows
for a substantial improvement in the accuracy of estimated soil characteristics (e.g. irrigation
need or carbon stocks within each SMU).

In 2017, a working group from the European Soils Bureau and University of Copenhagen
discussed the methodology for creating a level 2 SPADE database (SPADE 18). Given that it
took about 20 years to create the level 1 database, it was decided to speed up the process by
following the route used to finalise SPADE-14. The following concept has been developed based
on data from two member states, Denmark and UK.



1: For each country unique combinations of all soil types and topsoil textures present as
dominant, associated or included STUs were listed. For UK 79 new soil types had to be added to
the 62 at level 1, and for Denmark this left 29 unique combinations compared to 13 at level 1,
where only dominant soil types were considered. Thus, 16 new soil types had to be added to the
Danish database.

2: For each missing soil type, the entire level 1 database was scrutinised for the particular soil
type. If multiple countries contained the soil type, profiles from neighbouring countries had
preference. If more than one neighbouring country had the desired soil type, agricultural land use
had preference.

3: In cases where the soil type did not exist as a dominating soil type for any other country in the
database, the soil types were taken from a database containing modified soil profile data. This
database was created by compiling a list of all combinations of soil and topsoil texture in the
entire SPADE database that did not exist as dominating in any country, and therefore had no
estimated data assigned at level 1 (129 unique combinations in total). In the same way as
described for the dominating soil types, data were estimated for these profiles by making minor
modifications to existing profiles. For example, a Podzol with a topsoil texture class 2 (Po-2)
could be created from a slight modification of the topsoil particle size distribution for a Po-1.
Other characteristics affected by the change in soil texture were adjusted accordingly.

4: After completion, the level 2 database will be shared with national stakeholders for evaluation,
and changes can be made to any data not found to be valid or meaningful. All comments and



changes must be reported to the committee within a given period. If no responses are provided,
the proposed database will be published, but the stakeholders are always welcome to submit their
national change requests to JRC.

5: The final version will be published through JRC's European Soil Data Centre (ESDAC)
website (http://esdac.jrc.ec.europa.eu/).


**SPADE applications: Root zone capacity and soil organic matter stocks in Europe**
Earlier versions of the SPADE have been used to estimate soil organic C-stocks (European
Commission, 2005). More recently, it was used to map wheel load carrying capacity in Europe
(Schjønning et al. 2015).

*Root zone capacity to 100 cm*
To demonstrate the use of the complete SPADE level 1 database for a relevant soil property, we
calculated the plant available water for crops having an effective root depth of 100 cm (e.g.
barley), also called root zone capacity ($RZC_{100}$) (Figure 2). Crop production on soils with $RZC_{100}$
<50 mm in Northern Europe and <100 mm in Southern Europe is highly dependent on irrigation.
RZC was estimated from the following equation:

$$RZC_{100} = \sum_{i=100} (VWC_{100i} - VWC_{1500i}) \times D_i$$


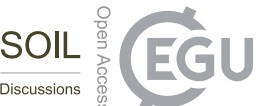

where $RZC_{100}$ is the cumulated root zone capacity (mm) within the upper 100 cm , $VWC_{1500i}$ is
the volumetric water content at 1500 kPa suction for horizon $i$ (%), $VWC_{100i}$ is the volumetric
water content at 100 kPa suction for horizon $i$ (%), and $D_i$ is the depth of horizon $i$ (mm).
Areas with very high $RZC_{100}$ (> 300 mm), relate mainly to the occurrence of Histosols, Gleysols
and Fluvisols, which are affected by shallow groundwater tables and few well-drained soils with
high silt and fine sand content (Figure 2). Soils with high $RZC_{100}$ are common in the Loess Belt,
just south of the ice margin from the previous ice ages, e.g. Belgium and Germany. The medium
$RZC_{100}$, 100-200 mm, corresponds mainly to loamy soils, for instance dominating in Eastern
Denmark, England and Poland, while sandy soils and some shallow loamy soils have a low
$RZC_{100}$ of 50-100 mm, e.g. Western Denmark and Sweden. Very shallow soils (Leptosols) have
a very low $RZC_{100}$ of 0-50 mm, which are found primarily in mountainous regions such as the
Alps, coastal Norway and large parts of Greece.



*SOC stock to 100 cm for Europe*
We estimated the SOC stock for Europe from the following equation:
$$SOC_{100} = \sum_{i=1} p_i \, SOC_i \, D_i \, A$$
where $SOC_{100}$ is the cumulated SOC stock to 100 cm depth, $p_i$ is the bulk density of horizon $i$,
$SOC_i$ is the SOC concentration for horizon $i$, $D_i$ is the depth of horizon $i$, and A is the area of the
particular STU (Figure 3). The regional distribution of soil organic C stocks is similar to what



was found previously (European Environmental Agency 2012; Panagos et al. 2013) with the
highest stocks concentrated in areas dominated by histosols (e.g. Northwestern British Isles and
Finland,  Figure 3). Intermediate stocks are situated in the wet Northwestern Iberian peninsula, in
the Massif Central region in France, and in the interior parts of the Scandinavian Peninsula,
while soils with relatively low SOC-stocks are situated in mountainous areas (e.g. coastal
Norway), dry Mediterranean areas, and areas under intensive cultivation (e.g. Northern France,
Germany, Denmark).
Our estimated cumulated SOC stock for Europe (0-100 cm) based on SPADE 18 level 1 is 76 x
$10^{15}$ g. This corresponds to the estimate of 75 x $10^{15}$ g obtained by the European Environment
Agency (2012) and the EC Joint Research Centre (Panagos et al. 2013) based on an earlier
version of the database, showing that our approach produces a similar result as obtained from
pedotransfer functions. We did not find other estimates of European SOC stocks across
landscape types in the scientific literature. However, as an approximation we may sum up the
recent estimates of SOC stocks in agricultural and forest soils. The forest SOC stock in Europe
(0-100 cm) was estimated to 22 x $10^{15}$ g (De Vos et al. 2015), while the agricultural SOC stock
(0-30 cm) was estimated to 18 x $10^{15}$ g (Lugato et al. 2014). This sums up to 40 x $10^{15}$ g SOC,
which is only about half of our SPADE 18 level 1 estimate. However, over-/underestimation of
~40-100% when comparing to other studies is similar to what was discovered by others (De Vos
et al. 2015; Guevara et al. 2018; Lugato et al. 2014). Hence, work still remains on elucidating the
underlying sources of variation to find the best approach, as estimates of SOC is considered an
important indicator of environmental health (European Environment Agency 2012; Panagos et
al. 2013).



*Better estimates with SPADE 18 level 2: SOC stock in Denmark*
The application of SPADE 18 level 2 data has been tested in a pilot study calculating the RZC
for wheat in Denmark (Jensen et al. 1998). They found a substantial difference of up to 49% in
estimated national RZC values when comparing level 1 to level 2 data. To demonstrate the added
value from including the associations and inclusions in another example, we calculated the soil
organic carbon stock (SOC) to 1 m depth for Denmark based on SPADE 14 level 1 (Figure 4a)
and SPADE 18 level 2 (Figure 4b) data.
Overall, the comparison shows that the estimated total SOC stock in the upper metre of Danish
soils increases by 14% from 478 x $10^{12}$ to 545 x $10^{12}$ g C when using level 2 data instead of level
1. This number is comparable to the most recent estimate obtained from digital soil mapping of
about 570 x $10^{12}$ g C (Adhikari et al. 2014) and previous estimates ranging from 563-598 x $10^{12}$
C (Krogh et al. 2003), suggesting that using level 2 data yields more reliable results. The
increase in SOC-stock using level 2 compared to level 1 data is mostly due to SOC-rich soils
such as Histosols, Gleysols and Fluvisols primarily present as associations or inclusions. The
spatial distribution of the changes reveals that particularly in areas dominated by loamy soils, the
inclusion of subordinate soil types increased the SOC stock substantially (Figure 4c),
occasionally more than 40%. For sandy soils, the carbon gain was modest, typically less than
20%. Only in small loamy areas in Western Jutland and on the raised seafloors in Northern
Jutland dominated by wetland soils, did the carbon content decrease by using the level 2
database, probably due to the inclusion of sandy soils with low organic matter content. This



study demonstrates the added accuracy of estimating an environmentally relevant soil property
like SOC stock by the more detailed level 2 database.



**Limitations of our approach**
Digital soil mapping (DSM, reviewed in Mulder et al. (2011); Minasny and McBratney (2016);
Zhang et al. 2017) is the future of soil mapping, and is constantly developing and improving (e.g.
Møller et al. 2019; Pouladi et al. 2019; Stockmann et al. 2015; Zeraatpisheh et al. 2019). The
great advantage of these formalised approaches are their reproducibility and ability to estimate
the accuracy of their predictions. However, as mentioned earlier, challenges to such inference
techniques persist, and no adequate and harmonised methodology for large-scale analyses has yet
been developed (Mulder et al. 2011; Zhang et al. 2017). Until these tools are developed, we
argue that databases with analytical soil properties estimated or evaluated by local expert
stakeholders is still a feasible way of assessing large-scale soil property patterns. Similar
conclusions underlie data harmonisation initiatives at the global scale lead by ISRIC, which has
led to the construction of the Global Soil Map (Arrouays et al. 2014) and the SoilGrids1km
(Hengl et al. 2014) related to the Harmonized World Soil Database (HWSD, Nachtergaele et al.
2014). The HWSD contains soil properties data gathered in various ways, resulting in
considerable variation in confidence levels. The European dataset for the HWSD is retrieved
from the European Soil Database, which comprises the information from the most recent SPADE
dataset (i.e. the one presented in this paper).



A consideration with respect to the interpretation of outputs from bottom-up harmonised
databases, like SPADE, is how well the mapping units actually reflect real landscape
delineations (Figure 1a). Efforts have been made by the ESDAC to let mapping units overlap
arbitrary administrative limits, such as national borders, to best fit the SMU delineations on both
sides (e.g. European Commission 2005). However, the inherent variation in level of detail from
the national datasets is still evident in certain areas (see for instance the Danish-German border).
Hence, the predictions based on the current dataset might be substantially improved by modern
downscaling techniques (as an example, see Peng et al. 2017 for a review of the downscaling of
soil moisture). This was beyond the scope of the current work, but should be a priority in future
large-scale soil mapping efforts.

## Conclusions

We document the development of a full-covered EU-wide soil database, containing analytical
data connected to the Soil Map of Europe at scale 1:1,000,000. We demonstrate the benefits of
careful analysis of legacy data, wherever possible with the help of national soil experts.
The application of the current soil analytical database at level 1 was demonstrated by calculating
the root zone capacity for the Europe and associated countries, mapping out areas where severe
need of irrigation for crop production might occur. Moreover, we estimate the SOC stock for
Europe to $76 \times 10^{15}$ g, which is larger than previous estimates. The increased accuracy obtained
by including associated and included soil types in the SPADE database, was demonstrated by
comparing the SOC stock of Denmark calculated from level 1 and level 2 data, showing an
increase of 14 % from $478 \times 10^{12}$ to $545 \times 10^{12}$ g C, which is more comparable to literature



estimates obtained with other methods. This exercise highlights the need for a level-2 database
for the entire European continent.
Perhaps the greatest contribution of this research to the management and protection of Europe's
soils is the harmonisation of detailed soil profile data, hitherto unavailable across regions, but
now connected to the latest soil mapping.

## Acknowledgements

We want to warmly thank our late colleague and friend, Professor Henrik Breuning-Madsen,
who passed during the preparation of this manuscript. He has been a key figure in moving
European soil science forward over more than three decades. This work was financially
supported by the European Union through the EC Joint Research Centre. We thank all national
stakeholders for their contributions to the development of the SPADE database. For a full list of
stakeholders we refer to ESDAC's homepage http://esdac.jrc.ec.europa.eu/.





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



**Table 1:** Average bulk densities calculated from the SPADE 2 database. The mean, standard
deviation and the number of observations (n) are shown.

| OM | Depth | Bulk Density | Std. dev. | n |
|---|---|---|---|---|
| % | cm | g cm$^{-3}$ | g cm$^{-3}$ | |
| 90-100 | | 0.1 | 0.13 | 165 |
| 80-90 | | 0.1 | 0.05 | 81 |
| 70-80 | | 0.2 | 0.11 | 64 |
| 60-70 | | 0.2 | 0.13 | 36 |
| 50-60 | | 0.3 | 0.13 | 25 |
| 40-50 | | 0.4 | 0.08 | 28 |
| 30-40 | | 0.4 | 0.17 | 19 |
| 20-30 | | 0.8 | 0.31 | 35 |
| 10-20 | | 1.0 | 0.72 | 176 |
| 5-10 | | 1.1-1.2 | n/a | n/a |
| <5 | 0-25 | 1.3 | 0.18 | 400 |
| | 25-50 | 1.4 | 0.18 | 726 |
| | 50-75 | 1.4 | 0.17 | 719 |
| | 75-100 | 1.5 | 0.14 | 468 |
| | >100 | 1.5 | 0.18 | 714 |






**Table 2:** The origin of SPADE data at the national level. *Original* shows the soil profiles to
which the stakeholders originally provided data; *Profiles from other countries* show the soil
profiles for which data was copy-pasted from a similar country; *Modified profiles* show the soil
profiles to which slight adjustments were made; *Level 1 Total* shows the total number of
dominating soil profiles; *Level 2 Total* shows the total number of profiles, when associated soil
types were included.

| Country | Original (SPADE 8) | Profiles from other countries (SPADE 14) | Modified profiles (SPADE 14) | Level 1 Total (SPADE14) | Level 2 Total (SPADE 18) |
|---|---|---|---|---|---|
| AL | 14 (AL 1-14) | 13 ( AL 15-27) | 3 (AL 28-30) | 30 | 49 |
| AT | 0 | 23 (AT 1-23) | 4 (AT 24-27) | 27 | 35 |
| BE | 42 (BE 1-42) | 14 (BE 43-56) | 0 | 56 | 74 |
| BG | 0 | 16 (BG 1-16) | 7 (BG 17-23) | 23 | 40 |
| CH | 28 (CH 1-28) | 2 (CH 29-30) | 7 (CH 31-37) | 37 | 51 |
| CZ | 0 | 19 (CZ 1-19) | 7 (CZ 20-26) | 26 | 73 |
| DE | 60 (DE 1-60) | 15 (DE 61-75) | 2 (DE 76-77) | 77 | 149 |
| DK | 13 (DK 1-13) | 0 | 0 | 13 | 29 |
| EE | 11 (EE 1-11) | 2 (EE 12-13) | 4 (EE 14-17) | 17 | 26 |
| ES | 26 (ES 1-26) | 15 (ES 27-41) | 8 (ES 42-49) | 49 | 65 |
| FI | 6 (FI 1-6) | 1 (FI 7) | 0 | 7 | 12 |
| FR | 118 (FR 1-118) | 35 (FR 119-153) | 22 (FR 154-175) | 175 | 230 |
| GB | 41 (GB 1-41) | 15 (GB 42-56) | 6 (GB 56-62) | 62 | 141 |
| GR | 10 (GR 1-10) | 15 (GR 11-25) | 4 (GR 26-29) | 29 | 66 |
| HU | 40 (HU 1-40) | 10 (HU 41-50) | 11 (HU 51-61) | 61 | 92 |
| IE | 18 (IE 1-18) | 4 (IE 19-22) | 3 (IE 23-25) | 25 | 44 |
| IT | 21 (IT 1-21) | 11 (IT 22-32) | 9 (IT 33-41) | 41 | 91 |
| LT | 0 | 20 (LT 1-20) | 8 (LT 21-28) | 28 | 52 |
| LU | 0 | 10 (LU 1-10) | 2 (LU 11-12) | 12 | 26 |
| LV | 26 (LV 1-26) | 0 | 0 | 26 | 39 |
| NL | 20 (NL1-20) | 12 (NL21-32) | 0 | 32 | 42 |
| NO | 15 (NO1-15) | 0 | 1 (NO16) | 16 | 23 |
| PL | 0 | 28 (PL1-28) | 12 (PL29-40) | 40 | 63 |
| PT | 18 (PT 1-18) | 10 (PT 19-28) | 4 (PT 29-32) | 32 | 66 |
| RO | 28 (RO 1-28) | 29 (RO 29-57) | 21 (RO 58-78) | 78 | 116 |
| SE | 0 | 9 (SE 1-9) | 3 (SE 10-12) | 12 | 21 |
| SI | 0 | 15 (SI 1-15) | 9 (SI 16-24) | 24 | 31 |
| SK | 17 (SK 1-17) | 6 (SK 18-23) | 1 (SK 24) | 24 | 73 |
| **Total** | **571 (31%)** | **349 (19%)** | **158 (9%)** | **1079 (59%)** | **1819 (100%)** |




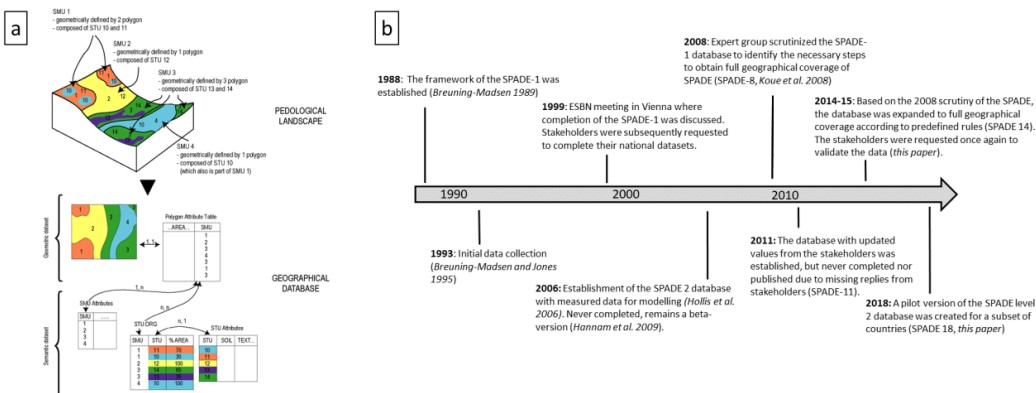


**Figure 1**:  a) Structure of the European Soil Database to which SPADE provides data (after
Lambert et al., 2003), b) Timeline of the establishment of the Soil Profile Analytical Database of
Europe (SPADE). See text for details.





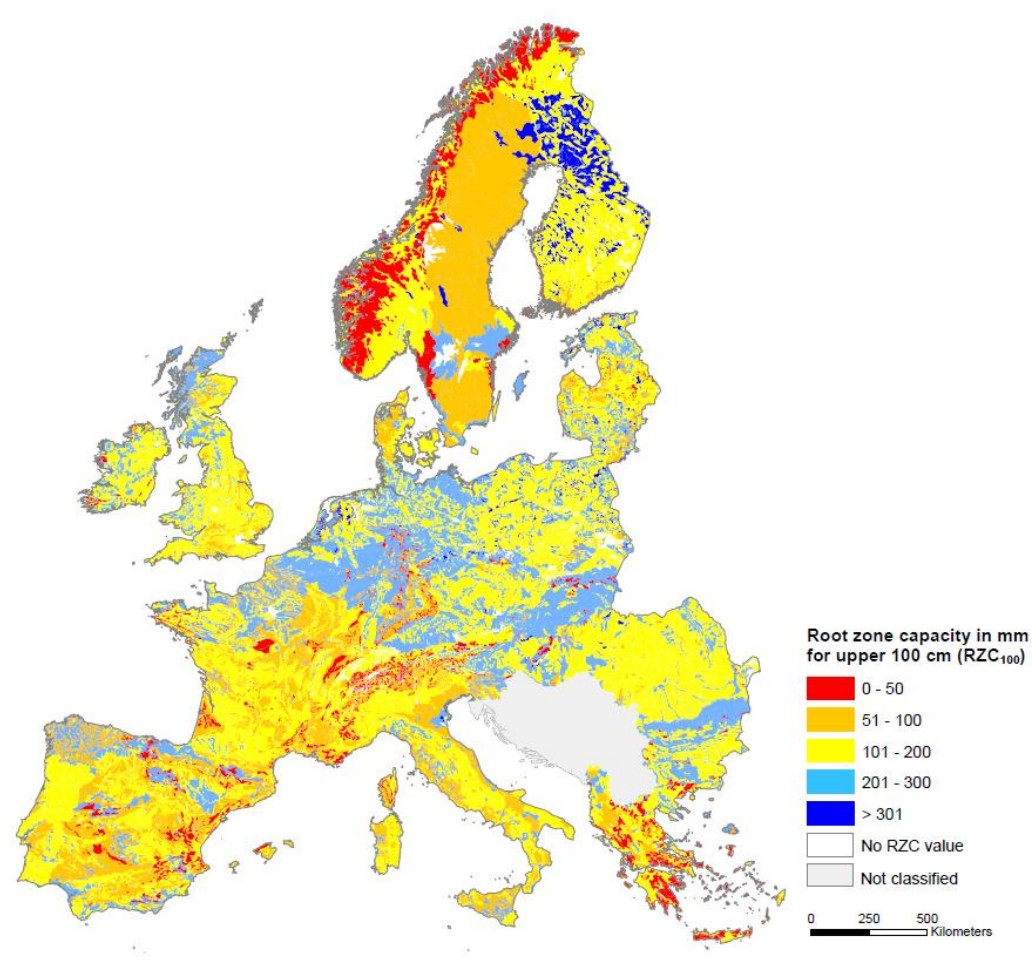


**Figure 2:** Plant available water content in mm within the uppermost one metre of the soil. Very

low 0-50 mm; low 50-100 mm, medium 100-200 mm; high 200-300 mm; very high >300

(mainly Histosol, Gleysol and Fluvisol).


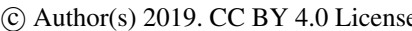



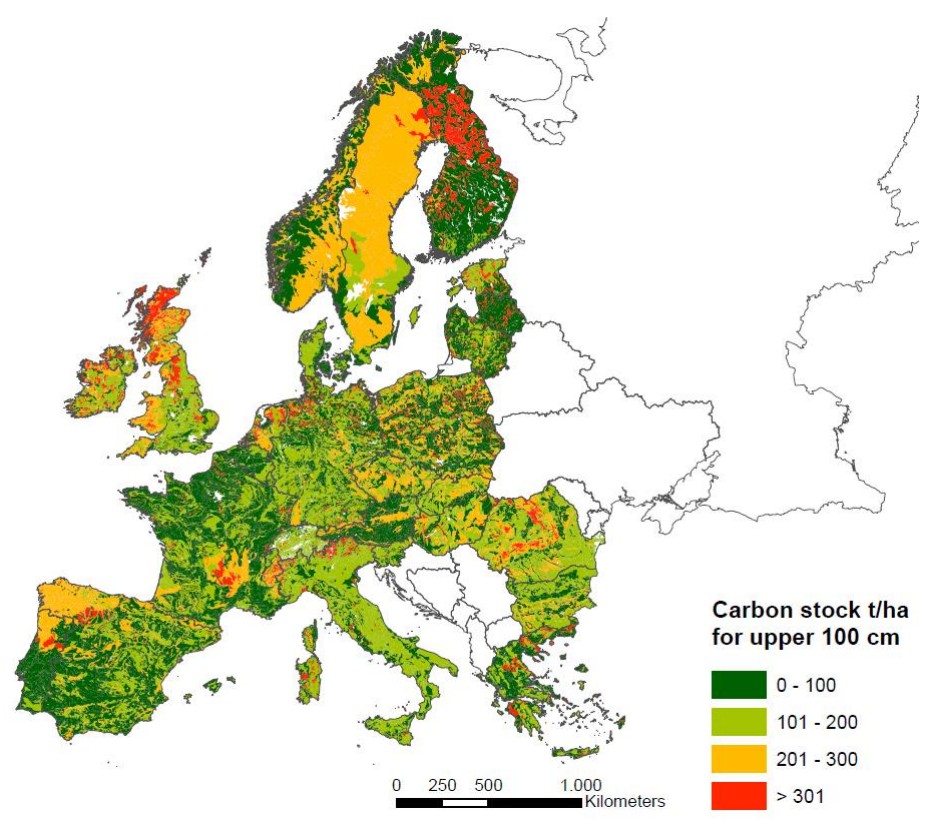


**Figure 3:** The soil organic carbon stocks (t ha$^{-1}$) in Europe within the upper 100 cm of soil

calculated based on level 1 data (dominating soil types only).






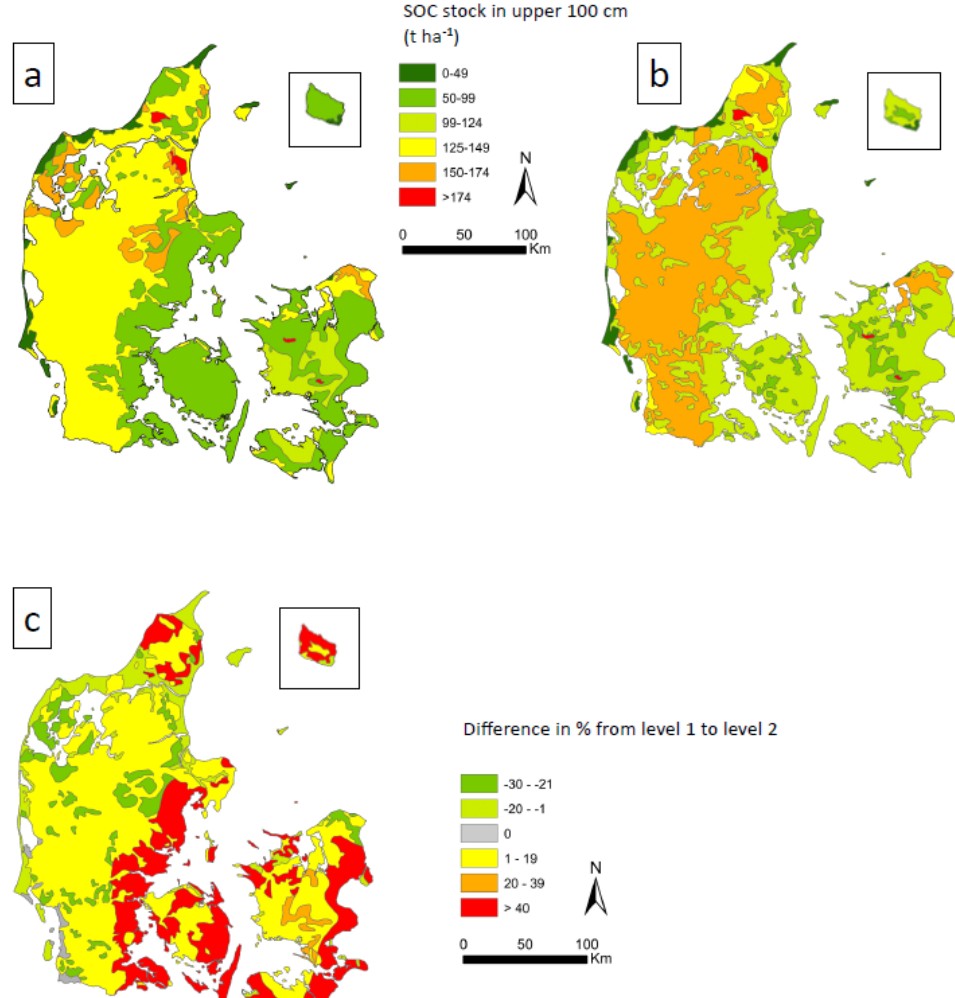


**Figure 4:** Soil organic carbon stocks (t ha-1) in Denmark within the upper 100 cm of the soil

calculated based on a) SPADE 18 level 1 data, and b) SPADE 18 level 2 data. c) Shows the

relative change from level 1 to level 2 in %.