# Peer review of "Title: Development of a harmonized soil profile analytical database for Europe: A"

_SOIL, 2019_

## Referee Comment (RC1) · Anonymous Referee #1 · 8 May 2019

This manuscript describes challenges associated with the development of a harmonised soil analytical database for Europe over the last 4-5 decades, with examples of use of the quality-assessed data upon their linkage to the 1:1M Soil Geographical Data Base of Europe.

Key issues of missing data, differences in soil analytical methods and their standardisation, and common lack of sharing of data for this significant European effort are duly discussed, reflecting commonly encountered issues in data compilation efforts of this broad scale nature.

The SPADE 18 database (see Fig. 2 and elsewhere, line 95-96) is currently under

development. Similarly, apparently, earlier versions 'remained as unpublished work in progress) (line 178-179). As such, the conclusions could be couched in terms of 'desirability of gaining free access (CC-BY) to profile data collected using public funds'.

The text can be tightened in places, i.e. remove the PTF regressions.

Other comments:

Abstract and elsewhere: Avoid using 'demonstrated', rather use shown or illustrated.

88: but (change to) → in which data from Europe are extracted from . . .

150: Hannam et al (2009) refers to an unpublished report. Should at least add the URL: https://esdac.jrc.ec.europa.eu/Esdb_Archive/eusoils_docs/esb_rr/SPADE-2_Beta_Report.pdf.

156-157: undertook a scrutiny → assessed the . . .

197: URL does not work. Similarly, the EU SPADE 14 database does not seem to be accessible (https://ec.europa.eu/knowledge4policy/dataset/jrc-esdac-114 and https://data.europa.eu/euodp/data/dataset/jrc-esdac-114 ), but its availability may be considered a prerequisite for publishing this manuscript. Similarly, the landing page for the dataset is non-operational (https://esdac.jrc.ec.europa.eu/content/spade-14) (08/05/2019).

199: 'stakeholder passivity', probably true, but should this be phrased as such in this manuscript? 200: The manuscript would benefit from a succinct description of these guidelines/or predefined equations.

211: 'before publication', according to the website these are 'provisional data' and the associated URL does not work (see above).

233: Add abbreviations for texture classes in text (as used in 242-249), e.g. <2um (TEXT2) etc. Alternatively, do these functions need to be defined here at all?

264: publishing SPADE-14 database. As indicated, not accessible online at the time of this review. 265-270: Table 2 is enlightening in the sense that it shows how few soil profiles were actually shared (1819) for consideration is SPADE over several decades.

274: The number of 1831 profiles for SPADE 18 is not consistent with Table 2 (1819). Based on a rough calculation, this would amount to some 0.4 profile per 1000 km2.

–: In my view, some discussion on 'data sharing', and desirability of open access (CC-BY) to profiles collated using public money, should be included in the discussions as a 'way forward'. See also: http://dx.doi.org/10.5194/essd-9-1-2017 and https://doi.org/10.1016/j.grj.2017.06.001. Possible synergies with the work of the GSP P4 & P5?

289: Please explain how this would lead to 'a substantial improvement in the accuracy of . . .'. How would this be quantified?

295: See comment. Database in preparation still?

342-354: This calculation gives a capacity, but does not consider whether there are any physical or chemical constraints for growth of specific crops, which would limit the effective 'capacity' (see e.g. https://doi.org/10.1016/j.geoderma.2018.02.046).

360: Commonly, a correction for the occurrence of coarse fragments (> 2mm) is considered in such calculations (https://www.soil-journal.net/3/61/2017/soil-3-61-2017.pdf). Is this the case for line 371-372.

396: This confirms the need to consider the full map unit (STMU) composition in such types of assessments.

417: Should add http://dx.doi.org/10.1371/journal.pone.0169748.

421: Actually, it has: http://dx.doi.org/10.1371/journal.pone.0169748.

243: At global level, using pedotransfer rules (interim update to HWSD), see http://dx.doi.org/10.1016/j.geoderma.2016.01.034

424-430: Not correct as written; should rephrase this. GSM and SoilGrids (now at 250m see above) are not related to the development of the HWSD, rather initiated in realisation of the need to improve on "conventional soil maps" using automated dsm procedures.

430: Not really possible as written. HWSD v1.2 was published in 2012. As such it cannot be based on the 'SPADE dataset described in this' manuscript.

444, 446, 450: replace demonstrated by shown or illustrated.

454: Alternatively, the increasing predictive capability and accuracy of digital soil mapping approaches should be indicated. Possibly, also make a reference to soil data collection/monitoring efforts such as LUCAS. Consideration of proximally derived soil data in future work other recent developments re. pedology-based and digital soil mapping (https://doi.org/10.1111/ejss.12790).

Figure 1. See 2018, SPADE 18 this paper. The dataset does not seem to be available from JRC ESDAC (https://esdac.jrc.ec.europa.eu/resource-type/soil-point-data); searching for 'SPADE 18' gives not results at al.

---

## Author Comment (AC1) · 25 Jun 2019

Development of a harmonized soil profile analytical database for Europe: A resource for supporting regional soil management

The authors are grateful for the comments provided on the manuscript by Anonymous Referee #1, to which we propose the following replies:

Comment 1: Abstract and elsewhere: Avoid using 'demonstrated', rather use shown or illustrated Response: noted and amended througout

Comment 2: 88: but (change to) → in which data from Europe are extracted from . . .

[Figure]

Response: done

Comment 3: 150: Hannam et al (2009) refers to an unpublished report. Should at least add the URL: Response: thank you for this. URL added to reference

Comment 4: 156-157: undertook a scrutiny → assessed the . . . Reponse: done

Comment 5: 197: URL does not work. Similarly, the EU SPADE 14 database does not seem to be accessible (https://ec.europa.eu/knowledge4policy/dataset/jrc-esdac-114 and https://data.europa.eu/euodp/data/dataset/jrc-esdac-114 ), but its availability may be considered a prerequisite for publishing this manuscript. Similarly, the landing page for the dataset is non-operational (https://esdac.jrc.ec.europa.eu/content/spade-14) Response: URL and landing page on ESDAC updated and now operational. Also available on EU Data Portral. URL in text modified to reflect access point in ESDAC.

Comment 6: 199: 'stakeholder passivity', probably true, but should this be phrased as such in this manuscript? Response: phrase removed

Comment 7: 200: The manuscript would benefit from a succinct description of these guidelines/or predefined equations. Response: Description of guidelines and equations are provided in subsequent text. For example, see 212 Comment 8: 211: 'before publication', according to the website these are 'provisional data' and the associated URL does not work (see above). Response: Final data now available through url

Comment 9: 233: Add abbreviations for texture classes in text (as used in 242-249), e.g. <2um (TEXT2) etc. Alternatively, do these functions need to be defined here at all? Response: abbreviations added

Comment 10: 264: publishing SPADE-14 database. As indicated, not accessible online at the time of this review. 265-270:. Response: Now accessible online

Comment 11: 274: The number of 1831 profiles for SPADE 18 is not consistent with Table 2 (1819). Based on a rough calculation, this would amount to some 0.4 profile per 1000 km2. 289: Please explain how this would lead to 'a substantial improvement

in the accuracy of . . .'. How would this be quantified? Response: sentence redrafted to remove the issue of accuracy

Comment 12: 295: See comment. Database in preparation still? Response: unclear. Full Level 2 database is still being developed. Is comment referring to work of GSP and open access? If so, see edits to conclusion.

Comment 13: 342-354: This calculation gives a capacity, but does not consider whether there are any physical or chemical constraints for growth of specific crops, which would limit the effective 'capacity' (see e.g. https://doi.org/10.1016/j.geoderma.2018.02.046). Response: No, physical and chemical constraints were not considered – this is simply an example to show how the SPADE database can be used, in this case just for the root zone capacity.

Comment 14: 360: Commonly, a correction for the occurrence of coarse fragments (> 2mm) is considered in such calculations (https://www.soil-journal.net/3/61/2017/soil-3-61-2017.pdf). Is this the case for line 371-372. Response: It was not, thanks for pointing this out. It is now corrected.

Comment 15: 396: This confirms the need to consider the full map unit (STMU) composition in such types of assessments. Response: Agree

Comment 16: 417: Should add http://dx.doi.org/10.1371/journal.pone.0169748. Response: Hengl 2017 added as reference.

Comment 17: 421: Actually, it has: http://dx.doi.org/10.1371/journal.pone.0169748. Response: Text amended

Comment 18: 243: At global level, using pedotransfer rules (interim update to HWSD), see http://dx.doi.org/10.1016/j.geoderma.2016.01.034 Response: We are not sue what is intended with this comment(?). Comment 19: 424-430: Not correct as written; should rephrase this. GSM and SoilGrids (now at 250m see above) are not related to the development of the HWSD, rather initiated in realisation of the need to improve on
"conventional soil maps" using automated dsm procedures. Response: Thanks for this clarification. Text rephrased.

Comment 20: 430: Not really possible as written. HWSD v1.2 was published in 2012. As such it cannot be based on the 'SPADE dataset described in this' manuscript. Response: Text amended to make reference to original HWSD

Comment 21: 444, 446, 450: replace demonstrated by shown or illustrated. Response: done

Comment 22: 454: Alternatively, the increasing predictive capability and accuracy of digital soil mapping approaches should be indicated. Possibly, also make a reference to soil data collection /monitoring efforts such as LUCAS. Consideration of proximally derived soil data in future work other recent developments re. pedology-based and digital soil mapping (https://doi.org/10.1111/ejss.12790). Response: Text added to recognise the contribution of LUCAS and precision farming. Reference to LUCAS Soil added.

Comment 23: Figure 1. See 2018, SPADE 18 this paper. The dataset does not seem to be available from JRC ESDAC (https://esdac.jrc.ec.europa.eu/resource-type/soil-point-data); searching for 'SPADE 18' gives not results at al. Response: Figure will be amended to show 2019 as this paper. Data now online. Comment 24: As such, the conclusions could be couched in terms of' desirability of gaining free access (CC-BY) to profile data collected using public funds'. –: In my view, some discussion on 'data sharing', and desirability of open access (CC-BY) to profiles collated using public money, should be included in the discussions as a 'way forward'. See also: http://dx.doi.org/10.5194/essd-9-1-2017 and https://doi.org/10.1016/j.grj.2017.06.001. Possible synergies with the work of the GSP P4 & P5? Response: New section added to conclusion addressing these issues

Comment 25: Remove the PTF regressions. Response: we prefer to maintain the regression equations in 242-250

---

## Author Comment (AC2) · 25 Jun 2019

We have attached the updated manuscript as a supplementary

Please also note the supplement to this comment:
https://www.soil-discuss.net/soil-2019-18/soil-2019-18-AC2-supplement.pdf
* * *

---

## Referee Comment (RC2) · Anonymous Referee #2 · 3 Jul 2019

MS: soil-2019-18 Title: Development of a harmonized soil profile analytical database for Europe: A 2 resource for supporting regional soil management

General comments This MS is informative on how SPADE was developed and evolved over the years. Unfortunately, the descriptions are not detailed enough so that potential users are convinced to use SPADE for their research endeavors. Many questions that raised my mind are mentioned below. The two examples for application of the SPADE 18 at EU level were not very convincing. Regarding the Root zone capacity I presume an error in the equation applied (hopefully it is just a typing error so that calculated results are all right) and for the SOC stock estimation no coarse fragments

are taken into account, not even mentioned. More references could have been made to recent papers and studies and to the applied methods. Over all, this MS can be still be improved substantially, starting by considering my suggestions and corrections seriously. I look forward to a next revision and a proper answer to my questions.

Specific comments L36-37. In the lack of systematic cross-European soil analysis schemes, ... Several cross and pan-European soil analysis schemes exist already for decades. Examples are the UN-ECE ICP Forests soil manual with sampling schemes for forest soils in the systematic 16x16 km Level I forest soil network. Freely accessible though this link. For agricultural soils mainly, a manual and sampling scheme was developed for the LUCAS monitoring grid. Reference: Fernández-Ugalde O., Orgiazzi A., Jones A., Lugato E., Panagos P., LUCAS 2018 – SOIL COMPONENT: Sampling Instructions for Surveyors, EUR 28501 EN, doi 10.2760/023673. Strange it is not mentioned in this manuscript, since different coauthors are involved. Also in World Reference Base, soil analytical methods required for soil description and classification are well described (see IUSS Working Group WRB (2014) among other FAO reports)

L55-57. A recent assessment ... This phrase is unclear and difficult to understand, unless you read the referenced article. Please rephrase

L64-L66. I expect some critical evaluation in this MS concerning the mapping-unit approach, especially the practical problems in GIS processing since soil-types are not spatially explicit defined this way. Maybe state some alternative (better) soil mapping approaches from literature.

L87-96. Overall, this section is not very clear in describing the various database products and how they are linked to each other, and what they effectively contain (which countries, number of soil profiles, measured vs. estimated/derived profile data, etc). I suggest to use Table 2 earlier in the text to describe the SPADE versions and their evolution.

L98. I would replace "Root zone capacity" here with "volumetric water content" since

the first term is not used frequently and should be defined first along with a proper reference.

L106. Why "preferable on arable land" ? Would have been more logical to provide analytical data for the dominant STU and for the dominant land-use in each SMU in Level 1, while in Level 3 you then have the differentiation among land-uses fo all soil-types.

L115. What do you exactly mean by "established analytical procedures" ? Do you refer to international procedures like ISO methods, EN, etc ? Or do you mean specific 'established manuals' ?

L115-116. I would expect the inverse: established procedures (understood as internationally accepted conventional methods) are more comparable across country borders than national methods (Proforma I).

L122-123. Why is the database limited to assessments of agricultural land management if one-third of the EU land area are forests ? Considering also that soil profiles under forest are well suited as a reference and to better evaluate the impact of agricultural management on soil development and quality.

L132-133. Can you provide any reason explaining the limited response from national stakeholders ? Was there any questionnaire or evaluation study dealing with this issue ?

L144. Was there any information on coarse fragments (stoniness) in each horizon ?

L149&L151. Versioning for SPADE databases is quite confusing. So you have SPADE2v11 (11th version of SPADE 2 ?), but also a SPADE version released in 2014, being SPADE14. I presume there is some simplification possible here ?

L167-168. If implausible values were adjusted, was this documented in the database. If so, how ?

L168-179. It seems that response of stakeholders for reviewing SPADE databases was repeatedly low or absent. Could it be that EC asked for responses and reviews on a voluntary basis, while only by (co) financing serious and adequate expert responses may be expected ? In this MS it seems that often the national stakeholders are blamed. Maybe the approach and strategy of responsible institutions like DG JRC is inadequate and does not promote fruitful cooperation between EC and member states on European soil databases?

L192. Matching of similar soil types in neighbouring countries is quite tricky. Thorough validation and evaluation of such a process is needed to avoid systematical bias in the SPADE database. Estimated records need a clear flag in the database so that they can be omitted by evaluators if they do not trust the estimated/imputed records.

L196. Replace 'final' by 'resulting'. Is there any further versioning of the SPADE14 database ?

L199. 'passivity' – There might be several reasons for not cooperating. See comment for L168-179

L201-L203. This all seems tricky to me. Is this process clearly documented and traceable ? Please inform the reader on this.

L206-208. Please provide a reference or URL to such a detailed description of the methodology

L210-211. Can you inform the reader how many stakeholders responded and how their response was processed before publication ?

L214 "weredeveloped" add space between words

L218. SPADE 2 (table 1), . . . please add "depending on their OM content and depth"

L223. BD estimated a value in the range 1.1-1.2 g cm-3. Why not using 1.15 g cm-3, making gap filling more reproducible ?

L229. Root zone capacities. Please refer to a definition or paper for this term (e.g. Jensen et al. 1998)

L233. Why using $50\mu$ in 20-50 $\mu$m and 50-200$\mu$m fractions while $63\mu$ is recommended by FAO and USDA soil texture classes ?

L234. "Complete estimated datasets" Does it mean completed using estimated data ? So, for instance estimated bulk density using PTF functions instead of measured bulk density ?

L240 VWC1000 not mentioned in line 231. VWC1500 in stead ?

L243 In all these equations BD is used twice, so this is a high impact predic tor. Is only measured BD applied here or also predicted BD ?

L250 please explain predictors in the regression equations: "where TEXT2 = 0-2 $\mu$m fraction in mass %, . . .

L261-262. Again, when adjusted, how is this documented in the database ?

L274-275. It would be helpful to provide an EU map showing the profile locations of SPADE 14 and SPADE 18 across Europe, so that the geographical distribution across Europe may be evaluated

L288. . . . assigned by estimated analytical data

L322 will be published. . . so without any national validation then. Is it indicated in the SPADE 18 database if the data has been nationally validated or not ?

L326 currently only the SPADE 14 can be downloaded through this link; and also SPADE/M and SPADE/M2 but these are not explained in this manuscript

L314& L345. Can you provide a reference for this equation ? It seems to me that VWC100i, which needs to be VWC at Field capacity is far too high at -100 kPa. Conventionally it should be for FC between -10, -20 or -32 kPa depending on soil texture,

respectively sand, silt or clayey soils. Can you check this ?

L350. I presume there is also a fraction 200-300mm considered "High"

L360. For the SOC stock estimation volumetric proportion of coarse fragments are not taken into account while these are usually considered for accurate SOC stock estimation (see De Vos et al. 2015 you are referring to) ? Why is this ? Because SPADE has no coarse fragments data ? Neglecting coarse fragments content will lead to an overestimation of the SOC stock.

L379-380. You cannot simply sum the estimates by De Vos et al. 2015 and Lugato et al. 2014 because the latter is only the 0-30 cm stock. According to the first referene, assuming a 60% proportion of SOC in the upper 30 cm, the total 1-m SOC stock in EU27 agricultural and forest soils would amount to ~51.3 Gt, which is about 68% of the SPADE 18 estimated stock. As said before, since coarse fragments are neglected in the calculation the SPADE 18 estimate is presumably an over-estimate. Recently the GSOC map was developed by the Global soil partnership. Please compare these results for Europe also with the SPADE 18 data, if necessary only for 0-30 cm topsoil SOC stocks.

L402. Indeed. This is a very important factor. Carbon hotspots are often smaller SMU's and often underrepresented in soil databases or masked by generalization of soil maps.

L436. Refer to figure please for Danish-German border example. . .

---

## Author Comment (AC3) · 20 Jul 2019

Development of a harmonized soil profile analytical database for Europe: A resource for supporting regional soil management

The authors are grateful for the comments provided on the manuscript by two anonymous referees, to which we propose the following replies:

Anonymous referee #1

Abstract and elsewhere: Avoid using 'demonstrated', rather use shown or illustrated – noted and amended throughout 88: but (change to) → in which data from Eu-

rope are extracted from . . . - done 150: Hannam et al (2009) refers to an unpublished report. Should at least add the URL: - thank you for this. URL added to reference 156-157: undertook a scrutiny → assessed the . . . - done Comment 5: 197: URL does not work. Similarly, the EU SPADE 14 database does not seem to be accessible (https://ec.europa.eu/knowledge4policy/dataset/jrc-esdac-114 and https://data.europa.eu/euodp/data/dataset/jrc-esdac-114 ), but its availability may be considered a prerequisite for publishing this manuscript. Similarly, the landing page for the dataset is non-operational (https://esdac.jrc.ec.europa.eu/content/spade-14)

Response: URL and landing page on ESDAC updated and now operational. Also available on EU Data Portral. URL in text modified to reflect access point in ESDAC.

Comment 6: 199: 'stakeholder passivity', probably true, but should this be phrased as such in this manuscript?

Response: phrase removed

Comment 7: 200: The manuscript would benefit from a succinct description of these guidelines/or predefined equations.

Response: Description of guidelines and equations are provided in subsequent text. For example, see 212

Comment 8: 211: 'before publication', according to the website these are 'provisional data' and the associated URL does not work (see above).

Response: Final data now available through url

Comment 9: 233: Add abbreviations for texture classes in text (as used in 242-249), e.g. <2um (TEXT2) etc. Alternatively, do these functions need to be defined here at all?

Response: abbreviations added

Comment 10: 264: publishing SPADE-14 database. As indicated, not accessible online

at the time of this review. 265-270:.

Response: Now accessible online

Comment 11: 274: The number of 1831 profiles for SPADE 18 is not consistent with Table 2 (1819). Based on a rough calculation, this would amount to some 0.4 profile per 1000 km2. 289: Please explain how this would lead to 'a substantial improvement in the accuracy of . . .'. How would this be quantified?

Response: sentence redrafted to remove the issue of accuracy

Comment 12: 295: See comment. Database in preparation still?

Response: unclear. Full Level 2 database is still being developed. Is comment referring to work of GSP and open access? If so, see edits to conclusion.

Comment 13: 342-354: This calculation gives a capacity, but does not consider whether there are any physical or chemical constraints for growth of specific crops, which would limit the effective 'capacity' (see e.g. https://doi.org/10.1016/j.geoderma.2018.02.046).

Response: No, physical and chemical constraints were not considered – this is simply an example to show how the SPADE database can be used, in this case just for the root zone capacity.

Comment 14: 360: Commonly, a correction for the occurrence of coarse fragments (> 2mm) is considered in such calculations (https://www.soil-journal.net/3/61/2017/soil-3-61-2017.pdf). Is this the case for line 371-372.

Response: It was not, thanks for pointing this out. It is now corrected.

Comment 15: 396: This confirms the need to consider the full map unit (STMU) composition in such types of assessments.

Response: Agree
Comment 16: 417: Should add http://dx.doi.org/10.1371/journal.pone.0169748.

Response: Hengl 2017 added as reference.

Comment 17: 421: Actually, it has: http://dx.doi.org/10.1371/journal.pone.0169748.

Response: Text amended

Comment 18: 243: At global level, using pedotransfer rules (interim update to HWSD), see http://dx.doi.org/10.1016/j.geoderma.2016.01.034

Response: The reference to Batjes (2016) is now added.

Comment 19: 424-430: Not correct as written; should rephrase this. GSM and Soil-Grids (now at 250m see above) are not related to the development of the HWSD, rather initiated in realisation of the need to improve on "conventional soil maps" using automated dsm procedures.

Response: Thanks for this clarification. Text rephrased.

Comment 20: 430: Not really possible as written. HWSD v1.2 was published in 2012. As such it cannot be based on the 'SPADE dataset described in this' manuscript. Response: Text amended to make reference to original HWSD

Comment 21: 444, 446, 450: replace demonstrated by shown or illustrated.

Response: done

Comment 22: 454: Alternatively, the increasing predictive capability and accuracy of digital soil mapping approaches should be indicated. Possibly, also make a reference to soil data collection /monitoring efforts such as LUCAS. Consideration of proximally derived soil data in future work other recent developments re. pedology-based and digital soil mapping (https://doi.org/10.1111/ejss.12790).

Response: Text added to recognise the contribution of LUCAS and precision farming. Reference to LUCAS Soil added.

[Figure]

Comment 23: Figure 1. See 2018, SPADE 18 this paper. The dataset does not seem to be available from JRC ESDAC (https://esdac.jrc.ec.europa.eu/resource-type/soil-point-data); searching for 'SPADE 18' gives not results at al.

Response: Data now online.

Comment 24: As such, the conclusions could be couched in terms of' desirability of gaining free access (CC-BY) to profile data collected using public funds'. – : In my view, some discussion on 'data sharing', and desirability of open access (CC-BY) to profiles collated using public money, should be included in the discussions as a 'way forward'. See also: http://dx.doi.org/10.5194/essd-9-1-2017 and https://doi.org/10.1016/j.grj.2017.06.001. Possible synergies with the work of the GSP P4 & P5?

Response: New section added to conclusion addressing these issues

Comment 25: Remove the PTF regressions.

Response: we prefer to maintain the regression equations in 242-250

Anonymous referee #2

General comments: This MS is informative on how SPADE was developed and evolved over the years. Unfortunately, the descriptions are not detailed enough so that potential users are convinced to use SPADE for their research endeavors. Many questions that raised my mind are mentioned below. The two examples for application of the SPADE 18 at EU level were not very convincing. Regarding the Root zone capacity I presume an error in the equation applied (hopefully it is just a typing error so that calculated results are all right) and for the SOC stock estimation no coarse fragments are taken into account, not even mentioned. More references could have been made to recent papers and studies and to the applied methods. Over all, this MS can be still be improved substantially, starting by considering my suggestions and corrections seriously. I look forward to a next revision and a proper answer to my questions.

Response: Thanks for the overall positive reception of our manuscript and for the helpful suggestions for improvements. We hope our answers are satisfactory.

Specific comments: Comment 1: L36-37. In the lack of systematic cross-European soil analysis schemes, . . . Several cross and pan-European soil analysis schemes exist already for decades. Examples are the UN-ECE ICP Forests soil manual with sampling schemes for forest soils in the systematic 16x16 km Level I forest soil network. Freely accessible though this link. For agricultural soils mainly, a manual and sampling scheme was developed for the LUCAS monitoring grid. Reference: Fernández-Ugalde O., Orgiazzi A., Jones A., Lugato E., Panagos P., LUCAS 2018 – SOIL COMPONENT: Sampling Instructions for Surveyors, EUR 28501 EN, doi 10.2760/023673. Strange it is not mentioned in this manuscript, since different coauthors are involved. Also in World Reference Base, soil analytical methods required for soil description and classification are well described (see IUSS Working Group WRB (2014) among other FAO reports)

Response: By "scheme" we meant "programme" or "plan" for actually going to the field and collecting soil samples systematically, rather than a formalised sampling protocol. We have changed "scheme" to "programme" to avoid this misunderstanding. Further, we have added a reference to the LUCAS soil collection programme, as suggested. Text has been amended in line 36-39 and 449-458.

Comment 2: L55-57. A recent assessment . . . This phrase is unclear and difficult to understand, unless you read the referenced article. Please rephrase

Response: Done

Comment 3: L64-L66. I expect some critical evaluation in this MS concerning the mapping-unit approach, especially the practical problems in GIS processing since soil-types are not spatially explicit defined this way. Maybe state some alternative (better) soil mapping approaches from literature.

Response: Considering the mapping scale (1:1 mio.) for Europe, it (still) makes good

sense to work with STUs grouped into SMUs containing soil associations and inclusions, because at that scale it would not be feasible to delineate the STUs. Therefore, the European Soil Database still uses the soil mapping unit (SMU) as the spatial unit on the soil map, but King et al.(1994) give a clear description of how this relates to soil types (STU). This work was done in the 'manual map drawing' era. However, without a massive new survey programme to increase the density of sampling points, it will not be possible to increase true spatial accuracy. To date, no Soil Survey organisation anywhere in Europe has provided sufficient resources to map the STUs spatially within each SMU. None of the DSM or pedometric approaches can overcome this problem. Thus, at the present time, the pedometric approach can only increase spurious spatial accuracy. Should the EU decide to improve the current soil map by taking it to a more detailed level, for example based on disaggregation of the existing soil by machine learning methods (see Møller et al. 2019), it might be appropriate to consider a cell-based data representation instead (similar to WISE30sec or SoilGrids), but that is, of course, to be decided among the experts involved. We have now added a discussion on this in the text (line 459-470).

Reference: Møller, A.B., Malone, B., Odgers, N.P., Beucher, A., Iversen, B.V., Greve, M.H. & Minasny, B. (2019): Improved disaggregation of conventional soil maps. Geoderma, 341, pp.148-160.

Comment 4: L87-96. Overall, this section is not very clear in describing the various database products and how they are linked to each other, and what they effectively contain (which countries, number of soil profiles, measured vs. estimated/derived profile data, etc). I suggest to use Table 2 earlier in the text to describe the SPADE versions and their evolution.

Response: The intension here was not to give an overview of the SPADE-databases, but rather why it is such a cornerstone in the ESDAC (other widely used databases extract data from it), and to introduce how this paper is structured. We have rephrased a bit to, hopefully, make this clearer.

Comment 5: L98. I would replace "Root zone capacity" here with "volumetric water content" since the first term is not used frequently and should be defined first along with a proper reference.

Response: We prefer to stick to root zone capacity, although we recognise that in certain soil-water disciplines other terms may be more common. We have now added a short explanation and some examples in line 246-7, and a reference to Jensen et al (1998).

Comment 6: L106. Why "preferable on arable land" ? Would have been more logical to provide analytical data for the dominant STU and for the dominant land-use in each SMU in Level 1, while in Level 3 you then have the differentiation among land-uses fo all soiltypes.

Response: When the principles were developed during the 80-90s, it was with the primary aim to provide data for modellers to solve agricultural problems. SPADE as the entire EUSIS was driven by the need to provide data for the agricultural crop forecasting system operated by MARS, known as Crop Growth Monitoring System (CGMS).

Comment 7: L115. What do you exactly mean by "established analytical procedures" ? Do you refer to international procedures like ISO methods, EN, etc ? Or do you mean specific 'established manuals' ?

Response: By "established" procedures, we mean analytical methods that are widely accepted, but not necessarily directly comparable. This could for example be CEC measured at different pH values. Therefore, Proforma I was established at the same time to have a dataset with estimated mean values based on the same methods. See Breuning-Madsen and Jones (1995) for further description of the standard methods. We have rephrased to make this clearer.

Comment 8: L115-116. I would expect the inverse: established procedures (understood as internationally accepted conventional methods) are more comparable across

country borders than national methods (Proforma I).

Response: See answer to Comment 8

Comment 9: L122-123. Why is the database limited to assessments of agricultural land management if one-third of the EU land area are forests ? Considering also that soil profiles under forest are well suited as a reference and to better evaluate the impact of agricultural management on soil development and quality.

Response: This should not be understood as if the database is limited to agricultural application, but rather that this was the primary objective at its establishment. We have now added a reference to Breuning-Madsen et al (1989), where the principles were originally defined.

Comment 10: L132-133. Can you provide any reason explaining the limited response from national stakeholders ? Was there any questionnaire or evaluation study dealing with this issue ?

Response: There was no questionnaire but the feedback we received was that essentially national soil survey organizations were lacking the resources to engage or that data were not available to third parties. We have added a short comment on this, and a section in the discussion (453-8).

Comment 11: L144. Was there any information on coarse fragments (stoniness) in each horizon ?

Response: Yes, there was. This has now been added to the text.

Comment 12: L149&L151. Versioning for SPADE databases is quite confusing. So you have SPADE2v11 (11th version of SPADE 2 ?), but also a SPADE version released in 2014, being SPADE14. I presume there is some simplification possible here ?

Response: We agree that the versioning of the SPADE databases are quite confusing and inconsistent – some were named after the year they were released, some after

their relative chronology, and some again after the year finalisation was initialised. We debated whether we should make a simpler nomenclature for the purpose of this publication, but decided that it might introduce further confusion. Instead, we decided to include the timeline in Figure 1 to hopefully make it the chronology clearer. We have added extra references to this figure in the text.

Comment 13: L167-168. If implausible values were adjusted, was this documented in the database. If so, how ?

Response: This was done with a standardised set of colour codes in the reports and country subsets of the database sent to the national stakeholders according to Koue et al (2008). From the stakeholders we corresponded with after we sent the evaluation reports and country-specific databases for approval, none of them mentioned any confusion related to this, so we believe our colour codes were clear and unambiguous.

Comment 14: L168-179. It seems that response of stakeholders for reviewing SPADE databases was repeatedly low or absent. Could it be that EC asked for responses and reviews on a voluntary basis, while only by (co) financing serious and adequate expert responses may be expected ? In this MS it seems that often the national stakeholders are blamed. Maybe the approach and strategy of responsible institutions like DG JRC is inadequate and does not promote fruitful cooperation between EC and member states on European soil databases?

Response: We do not intend any blame to the national stakeholders. We have now added some text on this (line 139, lines 453-8). Partly, as a consequence of the inadequate engagement by member states the EC recently set up its own data collection system (LUCAS SOIL COMPONENT), as you mentioned in comment 1.

Comment 15: L192. Matching of similar soil types in neighbouring countries is quite tricky. Thorough validation and evaluation of such a process is needed to avoid systematical bias in the SPADE database. Estimated records need a clear flag in the database so that they can be omitted by evaluators if they do not trust the estimated/imputed

records.

Response: The estimated values were clearly flagged in the databases sent to the national stakeholders for evaluation (see Breuning-Madsen et al. 2015).

Comment 16: L196. Replace 'final' by 'resulting'. Is there any further versioning of the SPADE14 database ?

Response: "Final" replaced by "Resulting". SPADE 14 is the final level 1 database, named after the year it was finalised (2014). SPADE 18 is the level 2 database, currently named after the year it was initiated (2018).

Comment 17: L199. 'passivity' – There might be several reasons for not cooperating. See comment for L168-179

Response: We have now added a small phrase on this. See response to comment 14.

Comment 18: L201-L203. This all seems tricky to me. Is this process clearly documented and traceable ? Please inform the reader on this.

Response: Yes, it is clearly traceable. Individual datasheets for each country documenting the process were sent to each stakeholder during the 2014-15 evaluation. An additional reference was added to Breuning-Madsen et al (2015) (line 204-5), where it is all described in detail.

Comment 19: L206-208. Please provide a reference or URL to such a detailed description of the methodology

Response: Reference added

Comment 20: L210-211. Can you inform the reader how many stakeholders responded and how their response was processed before publication ?

Response: We could, but it is probably not very informative about the engagement, as they were only requested to respond if they had any objections, questions or wanted to
change our suggested corrections.

Comment 21: L214 "weredeveloped" add space between words

Response: done

Comment 22: L218. SPADE 2 (table 1), . . . please add "depending on their OM content and depth"

Response: done

Comment 23: L223. BD estimated a value in the range 1.1-1.2 g cm-3. Why not using 1.15 g cm-3, making gap filling more reproducible ?

Response: In practice, we used 1.15 g cm3 unless the over-/underlying horizons gave us strong reason to believe it should differ. We have used a sentence stating that the OM range was also included in the assessment.

Comment 24: L229. Root zone capacities. Please refer to a definition or paper for this term (e.g. Jensen et al. 1998)

Response: See response to comment 5

Comment 25: L233. Why using $50\mu$ in 20-50 $\mu$m and 50-200$\mu$m fractions while $63\mu$ is recommended by FAO and USDA soil texture classes ?

Response: In the early 1970s, there were discussions at international level, mostly between American and European soil scientists on the definitions of soil particle sizes and how these relate to soil texture classes. The USDA texture classes were based on silt defined as 2-50um whereas the MIT size grade for silt used in civil-engineering is 2-63um. After in –depth discussion with the USDA and other European experts in the early 1970s, the Soil Survey of England & Wales (SSEW) adopted the MIT size grades for soil particle size and texture analysis. The relationship between 2-50um and 2-63um size fractions was developed for the conversion of historic data (Jones, 1975). But USDA still uses 2-50um as the particle size limits for silt and FAO only adopted

the 2-63um size limit for silt much later (ie FAO 2006, p27). For the SPADE databases therefore, the original 2-50um size grade for silt has been retained, also because much of the national soil data from European soil institutes was collected several decades ago.

References: FAO 2006. Guidelines for Soil Description, Fourth edition (97pp) – for particle size grades see p27. Hodgson, J M (ed) 1997. Soil survey field handbook. Soil Survey Technical Monograph No.5, Silsoe (116pp) - for particle size grades see p29. Jones, R J A. 1975. Soils in Staffordshire II, Soil Survey Record No.31 (158pp) – particle size conversion: [%(2-60um) = %(2-50um)+%(50-100um)0.26]. Schoeneberger, P J, Wysocki, D A, Benham, E.C. and Broderson, W D. 1998. Field book for describing and sampling of soils. Natural Resources Conservation Service, USDA, National Soil Survey Center, Lincoln, NE.

Comment 26: L234. "Complete estimated datasets" Does it mean completed using estimated data ? So, for instance estimated bulk density using PTF functions instead of measured bulk density ?

Response: "estimated" was erased, as it is indeed confusing. We meant complete datasets in the Proforma I database

Comment 27: L240 VWC1000 not mentioned in line 231. VWC1500 in stead ?

Response: Yes, thanks for finding this error. It is now corrected to 1500

Comment 28: L243 In all these equations BD is used twice, so this is a high impact predictor. Is only measured BD applied here or also predicted BD ?

Response: These are all measured values, as only countries with complete datasets were used to derive these equations.

Comment 29: L250 please explain predictors in the regression equations: "where TEXT2 = 0-2 $\mu$m fraction in mass %, . . .

Response: Yes, you are correct. We have now added further explanation.

Comment 30: L261-262. Again, when adjusted, how is this documented in the database ?

Response: These adjustments were also part of the colour coding system developed for the 2015-scrutiny (Breuning-Madsen et al. 2015).

Comment 31: L274-275. It would be helpful to provide an EU map showing the profile locations of SPADE 14 and SPADE 18 across Europe, so that the geographical distribution across Europe may be evaluated

Response: This is unfortunately not available due to the database structure with STUs and SMUs rather than individual soil profiles with coordinates.

Comment 32: L288. . . . assigned by estimated analytical data

Response: rephrased

Comment 33: L322 will be published. . . so without any national validation then. Is it indicated in the SPADE 18 database if the data has been nationally validated or not ?

Response: This has not yet been sent out for validation, but we will keep a record of this, once it is sent off.

Comment 34: L326 currently only the SPADE 14 can be downloaded through this link; and also SPADE/M and SPADE/M2 but these are not explained in this manuscript

Response: National stakeholders have not yet validated SPADE18, therefore, it is not publically available.

Comment 35: L314& L345. Can you provide a reference for this equation ? It seems to me that VWC100i, which needs to be VWC at Field capacity is far too high at -100 kPa. Conventionally it should be for FC between -10, -20 or -32 kPa depending on soil texture, respectively sand, silt or clayey soils. Can you check this ?

Response: Correct, this is a typo. FC should have been VWC10. It is now corrected. Thanks for pointing out this rather embarrassing mistake.

Comment 36: L350. I presume there is also a fraction 200-300mm considered "High"

Response: Correct, this is now added.

Comment 37: L360. For the SOC stock estimation volumetric proportion of coarse fragments are not taken into account while these are usually considered for accurate SOC stock estimation (see De Vos et al. 2015 you are referring to) ? Why is this ? Because SPADE has no coarse fragments data ? Neglecting coarse fragments content will lead to an overestimation of the SOC stock.

Response: We have now corrected the equation to subtract the coarse fragments before calculating the SOC. This reduce the estimate a bit, and with the adjustment of the estimate by Lugato et al (2014) you kindly suggest below, we are getting rather close to previous estimates of European SOC stocks. This has now been incorporated in the text.

Comment 38: L379-380. You cannot simply sum the estimates by De Vos et al. 2015 and Lugato et al. 2014 because the latter is only the 0-30 cm stock. According to the first referene, assuming a 60% proportion of SOC in the upper 30 cm, the total 1-m SOC stock in EU27 agricultural and forest soils would amount to 51.3 Gt, which is about 68% of the SPADE 18 estimated stock. As said before, since coarse fragments are neglected in the calculation the SPADE 18 estimate is presumably an over-estimate. Recently the GSOC map was developed by the Global soil partnership. Please compare these results for Europe also with the SPADE 18 data, if necessary only for 0-30 cm topsoil SOC stocks.

Response: Thanks for this excellent suggestion. See response to comment 37

Comment 39: L402. Indeed. This is a very important factor. Carbon hotspots are often smaller SMU's and often underrepresented in soil databases or masked by generalization of soil maps.

Response: Thanks. We agree, this is an often underappreciated point.

Comment 40: L436. Refer to figure please for Danish-German border example. . .

Response: Done

Please also note the supplement to this comment:
https://www.soil-discuss.net/soil-2019-18/soil-2019-18-AC3-supplement.pdf

---

## Author Comment (AC4) · 20 Jul 2019

- 1 **Title:** Development of a harmonized soil profile analytical database for Europe: A
- 2 resource for supporting regional soil management

**3 Authors:**

- 4 Jeppe Aagaard Kristensen1,2\*‡, Thomas Balstrøm2, Robert J.A. Jones3, Arwyn Jones4, Luca
- 5 Montanarella4, Panos Panagos4, and Henrik Breuning-Madsen2†‡.

**6 Affiliations:**

- 7 1Department of Physical Geography and Ecosystem Science, Lund University, Sölvegatan 12,
- 8 223 62 Lund, Sweden.
- 9 2Department of Geosciences and Natural Resource Management, University of Copenhagen,
- 10 1350 Copenhagen K, Denmark.
- 3School of Energy, Environment and AgriFood, Cranfield University, College Road, Cranfield,
- 12 MK43 0AL, UK.
- 4European Commission, DG Joint Research Centre, Via E. Fermi 2749, 21027 Ispra (VA), Italy.
- 14 *\*Correspondence to jeppe.aa.kristensen@gmail.com*
- 15  $^{\dagger}Deceased$
- 16 *‡**These authors contributed equally to this work.*
- 17 **Running head:** A harmonized soil profile analytical database for Europe.

**18 Abstract**

Soil mapping is an essential method to obtain a spatial overview of soil resources that are 19 increasingly threatened by environmental change and population pressure. Despite recent 20 21 advances in digital soil mapping techniques based on inference, such methods are still immature 22 for large-scale soil mapping. During the 1970s, 80s and 90s, soil scientists constructed a 23 harmonised soil map of Europe (1:1M) based on national soil maps. Despite this extraordinary 24 regional overview of the spatial distribution of European soil types, crude assumptions about soil properties were necessary to translate the maps into thematic information relevant for 25 management. To support modellers with analytical data connected to the soil map, the European 26 27 Soil Bureau commissioned the development of the Soil Profile Analytical Database for Europe 28 (SPADE) in the late 1980s. This database contains soil analytical data based on a standardised set of soil analytical methods across the European countries. Here, we review the principles 29 adopted for developing the SPADE database during the past five decades, and the work towards 30 fulfilling the milestones of full geographic coverage for dominant soils in all the European 31 32 countries (SPADE level 1), and the addition of secondary soil types (SPADE level 2). We illustrate the application of the database by showing the distribution of the root zone capacity, 33 and by estimating the soil organic carbon (SOC) stocks to a depth of 1 m for Europe to  $60 \times 10^{15}$ 34 35 g. The increased accuracy, potentially obtained by including secondary soil types (level 2), is shown in a case study to estimate SOC stocks in Denmark. Until data from systematic cross-36 European soil sampling programmes have sufficient spatial coverage for reliable data 37 interpolation, integrating national soil maps and locally assessed analytical data into a 38 harmonised database remains a powerful resource to support soil resources management at 39

40 regional and continental scales by providing a platform to guide sustainable soil management41 and food production.

42

Keywords: EU soil map; SPADE; Soil data harmonisation; Soil organic carbon; Root zone
capacity

45

**46 Introduction**

In a world subject to constant environmental change and increasing population pressure, soil 47 becomes an increasingly important but threatened resource (FAO 2015; Sustainable Food Trust 48 49 2015). This challenge must be met at multiple management levels and spatial scales; hence, accurate understanding of the available resources at the appropriate scale is required (e.g. 50 Robinson et al. 2017). In spite of advances in digital soil mapping using remote sensing and 51 geographical information systems to infer soil properties (McBratney et al. 2003; Arrouays et al. 52 2014; Minasny and McBratney 2016; Zhang et al. 2017), data and standardised methods for large 53 scale mapping are still inadequate. In particular, the existing methods are challenged in densely 54 vegetated areas and for subsoil properties (Mulder et al. 2011), which are highly relevant for 55 environmental management and food production. This was recently emphasised by the 56 57 suggesting that the uncertainty in soil data could potentially offset climate change impacts on future crop yields, due to the strong climate response dependence on soil type (Folberth et al. 58 2016). This notion calls for continued efforts to improve soil maps. 59

[revised manuscript text omitted]

Initially, two soil analytical databases were established; one containing estimated mean values 116 117 for typical soil profiles according to a fixed set of standardised soil analytical procedures provided by national stakeholders (referred to as Proforma I), while another contained soil 118 profile data measured using established yet not necessarily cross-country standardised analytical 119 120 procedures (referred to as Proforma II). Thus, the Proforma I database contains data comparable 121 across country borders while this is not always the case for the Proforma II database (Breuning-Madsen and Jones 1995). In order to make the database functional as soon as possible for the 122 entire coverage area, each Member State stakeholder was asked to deliver one full set of 123 Proforma I (estimated) analytical data for each dominant soil type (STU) in each of the SMUs 124 125 delineated on the Soil Map of Europe (1:1M). Providing data for the Proforma II (measured) database was made optional to smooth the data collection procedure. Where possible, the data 126 127 should be provided for agricultural land, as the primary aim of the database was to underpin

128 large-scale assessments of agricultural land management (Breuning-Madsen et al. 1989; Vossen129 1993).

In 1993, Proforma I and II schemes (including guidelines) were sent to the stakeholders in order
to collect data for the individual countries; detailed guidelines for the compilation of the SPADE
1 dataset was published by Breuning-Madsen and Jones (1995).

Subsequently, the SPADE 1 database was expanded to include data from the new EU Member 133 134 States but also from non-EU European nations such as Albania, Norway and Switzerland. By the end of the 1990s, SPADE 1 was subject to a data quality assessment and scrutinised to identify 135 136 missing data and evaluate overall data reliability. Based on the recommendations presented at a European Soil Bureau Network (ESBN) meeting in Vienna 1999, the national stakeholders were 137 138 requested to update their individual datasets. Meanwhile, only a few national stakeholders 139 engaged in this exercise due to lack of resources or limitations on data dissemination, which left the SPADE 1 incomplete and not well suited for modelling at the European level. 140

141

**142 An attempt to populate SPADE with measured data (SPADE 2)**

Due to the limitations of SPADE-1, SPADE-2 was developed to derive appropriate soil profile
data to support, for example, higher tier modelling of pesticide fate at the European level (Hollis
et al., 2006). Data were supplied from national data archives, similar to SPADE 1 Proforma II.
Despite the analytical methods differing between countries, the raw national data were
harmonised and validated to provide a single data file for use in conjunction with the existing
Soil Geographical Data Base of Europe (Platou et al. 1989). The primary soil properties required
for each soil were: Horizon nomenclature (e.g. A, E, B, C), upper and lower horizon depth

150 (cm), particle-size distribution: clay, silt, total sand and content of at least 3 sand fractions, content of coarse fragments (>2 mm), pH in water (1:2.5 soil:water), organic carbon content (%) 151 and dry bulk density ( $g \text{ cm}^{-3}$ ). 152 The acquisition of data happened in two steps; first datasets were obtained from Belgium, 153 Luxembourg, Denmark, England and Wales, Finland, Germany, Italy, the Netherlands, Portugal 154 155 and Scotland (Hollis et al. 2006), and next the database was expanded with data from Bulgaria, Estonia, France, Hungary, Ireland, Romania, Slovakia, Spain, France and Ireland . Due to the 156 lack of methodological consistency between countries, the final database (SPADE2v11) was 157 158 never published, hence only exists as a beta version of collated datasets from the first and second phases of soil profile data acquisition (Hannam et al. 2009). However, it was used to 159 estimate bulk densities for missing data in the later SPADE 14 (see Figure 1b for timeline and 160 overview of the SPADE versions). 161

162

**163 Steps towards full geographical coverage (SPADE 8)**

In an effort to obtain a functional database with full spatial coverage for Europe, a small
specialist group from Denmark (Prof. Henrik Breuning-Madsen, Assoc. Prof. Thomas Balstrøm
and M.Sc. Mads Koue from the Institute of Geography, University of Copenhagen) assessed the
national datasets in 2008 using error finding equations based on literature values, expert
judgements, and pedotransfer functions (Koue et al. 2008).

169 First, a quality check was conducted on all data. This process consisted of:

i) cross-checking of interdependent variables (e.g. pH vs. base saturation or porosity vs.
saturated water content); and

ii) checking the plausibility of all values according to published theoretical or empirical
values (e.g. for bulk density (BS) or C:N-values).

Examples of common questionable data were occurrences of bulk soil C:N values <5, 174 mismatches between BS and pH (e.g. BS>90% at pH<4.5), and volumetric water content at 175 176 saturation exceeding the porosity. Based on this examination, implausible values were either adjusted to plausible values or marked as unlikely based on predefined criteria. All changes and 177 suggestions were carefully flagged to make them obvious to national evaluators. However, in 178 terms of spatial extent, it was still only possible to link a soil analytical dataset for a dominant 179 soil type to approximately 70% of the SMUs in the area covered by the database. 180 At an ESBN meeting in Paris, December 2008, the reviewed SPADE 8 database was discussed 181 182 and following the meeting, the national evaluation reports and the country specific databases were sent to the national stakeholders with a request to i) review and change the existing data to 183 plausible values based on the expert scrutiny, and ii) estimate new datasets for the dominant soil 184 185 types without data based on their local expertise. The modifications received from the stakeholders were incorporated in the SPADE 8 database that was renamed SPADE 11. 186 However, once again the data received from national stakeholders was inadequate, which still 187 left the database incomplete, so SPADE 11 remained as unpublished work in progress. 188

189

Figure 1. a) Structure of the database, b) Timeline showing the development of the database.

192 Establishing a SPADE for dominant soil types with full coverage of the EU (SPADE 14)

| 193 | Without further input from the national stakeholders, implausible data identified in SPADE 8       |
|-----|----------------------------------------------------------------------------------------------------|
| 194 | were estimated to make the Proforma I (level 1) database more functional for modelling. Thus,      |
| 195 | starting in 2014, the SPADE 8 database was updated by a working group consisting of the            |
| 196 | authors of the current paper.                                                                      |
| 197 | Specifically, this work package had three key goals:                                               |
| 198 | i: To implement the suggested improvements of the existing data in the SPADE database              |
| 199 | suggested during the 2008 evaluation,                                                              |
| 200 | ii: To estimate values for the profiles lacking data (approximately 32% of the dominant            |
| 201 | STUs) based on matching of similar soil types in neighbouring countries, the data in               |
| 202 | SPADE 2, or other reference data sources.                                                          |
| 203 | iii: To update the existing SPADE database with the complete dataset after revision by             |
| 204 | the national stakeholders.                                                                         |
| 205 | The resulting SPADE14 database is publically available through JRC's European Soil Data            |
| 206 | Centre (ESDAC) website ( https://esdac.jrc.ec.europa.eu/content/spade-14 ).                 |
| 207 | Firstly, the questionable values identified in SPADE 8, but not corrected by stakeholders, were    |
| 208 | adjusted to fit theoretical or average values according to predefined equations or guidelines (see |
| 209 | below and Breuning-Madsen et al. 2015). Secondly, data for profiles lacking stakeholder            |
| 210 | estimated values were assigned by copying complete datasets from identical soil types in           |
| 211 | neighbouring countries. If no matching profiles were identified, the search was extended to the    |
| 212 | entire database. Thirdly, data for the remaining ~15% of the dominant soil types (STUs for         |
| 213 | which no estimated data existed anywhere in the database) was created by adjusting existing data   |

214 from similar soil profiles, preferably from the country itself or neighbouring countries to minimise variation due to climate and parent material. The evaluation guidelines sent to the 215 stakeholders during the SPADE 14 evaluation provided a detailed description of the 216 217 methodology, and an overview of all modifications made with the suggested changes properly flagged with colour coding of adjusted values depending on the nature of the change (Breuning-218 219 Madsen et al. 2015). The entire database was quality controlled with the updated versions of equations and guidelines used during the 2008 evaluation thus ensuring consistency across 220 Member States. Finally, the quality controlled national data where sent to each stakeholder for 221 222 final checking and revision. The changes suggested by stakeholders were incorporated before publication. 223

224

**225 Examples of correction guidelines**

For some parameters, no correction guidelines were specified during the 2008 evaluation, in
which case they were developed during the 2014/15-evaluation. As examples, the estimation of
bulk density and volumetric water content are elaborated below.

229

**230 Bulk density**

[revised manuscript text omitted]

**305 Creating a pilot version of the SPADE 18 level 2 database (SPADE-18)**

As described previously, the SPADE framework has four levels. The level 2 database contains the same type of analytical data as the level 1 database, but in addition to the dominating soil types, the inclusions and associations have been assigned a set of estimated analytical data. This improves the use of the SGDBE to predict soil characteristics (e.g. irrigation need or carbon stocks) as users can assign values for all soil types within each SMU.

311

In 2017, a working group from the European Soils Bureau and University of Copenhagen discussed the methodology for creating a level 2 SPADE database (SPADE 18). Given that it took about 20 years to create the level 1 database, it was decided to speed up the process by following the route used to finalise SPADE-14, to have a complete dataset that could be subsequently improved by national stakeholders. The following concept were developed based on the work on finalising level 2 datasets from two member states, Denmark and UK.

[revised manuscript text omitted]

**375 Figure 2 EU RZC**

376

377 SOC stock to 100 cm for Europe

378 We estimated the SOC stock for Europe from the following equation:

379
$$SOC_{100} = \sum_{i=1}^{N} (1 - g_i) p_i SOC_i D_i A$$

where  $SOC_{100}$  is the cumulated SOC stock to 100 cm depth,  $g_i$  is the coarse particle fraction of horizon *i*,  $p_i$  is the fine earth (<2 mm) bulk density of horizon *i*,  $SOC_i$  is the SOC concentration for horizon *i*,  $D_i$  is the depth of horizon *i*, and A is the area of the particular STU, i.e. the area of the SMU multiplied by the proportional area covered by the STU (Figure 3). The regional distribution of soil organic C stocks is similar to what was found previously (European Environmental Agency 2012; Panagos et al. 2013). The highest stocks are concentrated in areas dominated by histosols (e.g. Northwestern British Isles and Finland, Figure 3). Intermediate
stocks are situated in the wet Northwestern Iberian peninsula, in the Massif Central region in
France, and in the interior parts of the Scandinavian Peninsula, while soils with relatively low
SOC-stocks are situated in mountainous areas (e.g. coastal Norway), dry Mediterranean areas,

and areas under intensive cultivation (e.g. Northern France, Germany, Denmark).

Our estimated cumulated SOC stock for Europe (0-100 cm) based on SPADE 14 (level 1) is 60 x 391  $10^{15}$  g. This compares to the estimate of 75 x  $10^{15}$  g obtained by the European Environment 392 Agency (2012) and the EC Joint Research Centre (Panagos et al. 2013) based on an earlier 393 version of the database, showing that our approach produces a somewhat lower result. We did 394 not find other estimates of European SOC stocks across landscape types in the scientific 395 396 literature. However, as an approximation we may sum up the recent estimates of SOC stocks in agricultural and forest soils. The forest SOC stock in Europe (0-100 cm) was estimated to 22 x 397  $10^{15}$  g (De Vos et al. 2015), while the agricultural SOC stock (0-30 cm) was estimated to 18 x 398  $10^{15}$  g (Lugato et al. 2014). As an attempt to roughly correct for the agricultural estimate only 399 covering the upper 30 cm of the soil profile, we assumed that the topsoil (0-30 cm) contained 400 401 about 60 % of the SOC stock in the top 100 cm (De Vos et al. 2015). Using this correction the estimate for the agricultural soils to 100 cm increased to  $30 \times 10^{15}$  g, so the estimates sum up to 402  $52 \times 10^{15}$  g SOC, which is quite similar to our SPADE 14 (level 1) estimate. Particularly 403 considering that over-/underestimation of ~40-100% when comparing to other studies are 404 common (De Vos et al. 2015; Guevara et al. 2018; Lugato et al. 2014). Nonetheless, work still 405 406 remains on elucidating the underlying sources of variation to find the best approach, as estimates of SOC is considered an important indicator of environmental health (European Environment 407 Agency 2012; Panagos et al. 2013). 408

- 410 Figure 3
- 411

**412 Better estimates with SPADE level 2: the SOC stock in Denmark**

The application of SPADE level 2 (SPADE18) data has been tested in a pilot study calculating the RZC for wheat in Denmark (Jensen et al. 1998). They found a substantial difference of up to ~50% in estimated national RZC values when comparing level 1 to level 2 data. To show the added value from including the associations and inclusions in another example, we calculated the soil organic carbon stock (SOC) to 1 m depth for Denmark based on SPADE 14 (level 1, Figure 4a) and SPADE 18 (level 2, Figure 4b) data.

419 Overall, the comparison shows that the estimated total SOC stock in the upper metre of Danish soils increases by 12% from 332 x  $10^{12}$  to 378 x  $10^{12}$  g C when using level 2 data instead of level 420 1. This number is higher, yet not quite as high as the most recent estimate obtained from digital 421 soil mapping of about 570 x  $10^{12}$  g C (Adhikari et al. 2014) and previous estimates ranging from 422 563-598 x 1012 C (Krogh et al. 2003), but it suggests that using level 2 data yields more 423 comparable results than using level 1. The increase in SOC-stock using level 2 compared to level 424 425 1 data is mostly due to SOC-rich soils such as Histosols, Gleysols and Fluvisols primarily present as associations or inclusions. The spatial distribution of the changes reveals that 426 427 particularly in Northern Jutland on the raised seabeds, the inclusion of subordinate soil types 428 increased the SOC stock substantially (Figure 4c), occasionally more than 30% (red areas). For 429 sandy soils (Western Jutland), the carbon gain was modest, typically less than 20%. Only in 430 small loamy SMUs in Western Jutland did the carbon content decrease by using the level 2

database, probably due to the inclusion of sandy soils with relatively low organic matter content.
This study highlights the added accuracy of estimating an environmentally relevant soil property
like SOC stock by the more detailed level 2 database, which yielded estimates more similar to
the estimates obtained with pedometric (Krogh et al. 2003) and advanced interpolation
approaches (Adhikari et al. 2014) than results based on SPADE level 1.

436

**437 FIGURE 4**

438

**439 Limitations of our approach**

Digital soil mapping (DSM, reviewed in Mulder et al. 2011; Minasny and McBratney 2016; 440 441 Zhang et al. 2017) is the future of soil mapping, and is constantly developing and improving (e.g. Hengl et al 2017, Møller et al. 2019; Pouladi et al. 2019; Stockmann et al. 2015; Zeraatpisheh et 442 al. 2019). The great advantage of these formalised approaches are their reproducibility and 443 ability to estimate the accuracy of their predictions. However, as mentioned earlier, challenges to 444 445 such inference techniques persist (Mulder et al. 2011; Zhang et al. 2017), particularly data scarcity is a major challenge. Similar conclusions underlie data harmonisation initiatives at the 446 global scale lead by ISRIC, which has led to the construction of the Global Soil Map (Arrouays 447 448 et al. 2014), the SoilGrids (Hengl et al. 2014; 2017), the Harmonized World Soil Database (HWSD, Nachtergaele et al. 2014) and the WISE30sec (Batjes 2016). To overcome this, the EU 449 450 recently launched the LUCAS 2018 - SOIL COMPONENT (Fernández-Ugalde et al. 2017), 451 which is a soil sampling programme that will provide measured soil data from ~27,000 profiles 452 covering the European area.

However, to supplement such approaches until data availability increases, databases with
analytical soil properties estimated or evaluated by local expert stakeholders are still a feasible
way of assessing large-scale soil property patterns, which is substantiated by our ability to
estimate similar distributions and stocks as previous studies. Yet, our voluntary approach is
vulnerable to inadequate stakeholder engagement, which has been a challenge throughout this
process. This adds to the justifications of the LUCAS 2018 – SOIL COMPONENT.

A consideration with respect to the interpretation of outputs from bottom-up harmonised 459 databases, like SPADE, is how well the mapping units actually reflect real soil and landscape 460 delineations (Figure 1a). Efforts have been made by the ESDAC to let mapping units overlap 461 462 arbitrary administrative limits, such as national borders, to best fit the SMU delineations on both 463 sides (e.g. European Commission 2005). However, the inherent variation in level of detail from the national datasets is still evident in certain areas (see for instance the Danish-German border 464 465 in maps in European Commission (2005)). Therefore, the predictions based on the current dataset might be improved by modern downscaling techniques (see Møller et al. 2019 for an 466 example), but it might be appropriate to consider a cell-based data representation if further 467 disaggregation was to be implemented. However, considering the scale of the EU soil map 468 (1:1M), it is not feasible to delineate single STUs, so working with SMUs with a set of STUs is 469 still feasible for this purpose. 470

471

**472 Concluding remarks**

We document the development of a full-covered EU-wide soil database, containing analytical
data connected to the Soil Map of Europe at scale 1:1,000,000. We show the benefits of careful
analysis of legacy data, wherever possible with the help of national soil experts.

The application of the current soil analytical database at level 1 was illustrated by calculating the 476 root zone capacity to 100 cm for the Europe and associated countries, mapping out areas where 477 severe need of irrigation for crop production might occur. Moreover, we estimate the SOC stock 478 to 100 cm for Europe to  $60 \times 10^{15}$  g, which is comparable to previous estimates. The increased 479 accuracy obtained by including associated and included soil types in the SPADE database, was 480 presented by comparing the SOC stock of Denmark calculated from level 1 and level 2 data, 481 showing an increase of 12 % from 332 x  $10^{12}$  to 378 x  $10^{12}$  g C, which is closer to literature 482 estimates obtained with other methods. This exercise highlights the need for a level-2 database 483 484 for the entire European area.

Perhaps the greatest contribution of this research to the management and protection of Europe's 485 486 soils is the harmonisation of detailed soil profile data, hitherto unavailable across regions, but now connected to the latest soil mapping. These considerations are driving initiatives such as the 487 soil component of the LUCAS survey, which by generating harmonised and comparable data on 488 topsoil characteristics across the EU (Orgiazzi et al. 2014) is increasing the predictive capability 489 and accuracy of digital soil mapping approaches. In time, soil mapping will need to 490 491 accommodate high data streams that will be driven by precision farming, proximal sensing and the Internet of Things (Carolan 2017), but until sufficient data amounts exist, databases with 492 expert estimated data like the current SPADE is a good supplement. 493

494 Finally, while soils are often under land in private ownership, there is the increasing recognition of soil as a 'public good' that provides society with key ecosystem services. In such a paradigm, 495 there is a strong case to be made for providing unrestricted access to soil data. Many national soil 496 institutions regard soil profiles as 'primary data sources' that underpin revenue earning systems. 497 However, there is a strong case for inherent soil data (i.e. texture, carbon, pH, nutrient content, 498 499 CEC, EC, etc.) that reflect pedogenic processes and basic land management practices to be publically available (with appropriate attribution or data sharing licence). Such an approach, 500 501 possibly driven by the aims of the Global Soil Partnership to enhance the quantity and quality of 502 soil data and data collection, could lead to a more rapid completion of the higher-level orders of SPADE, while at the same time provide new understanding in pedogenesis and the need for 503 further research. 504

505

**506 Acknowledgements**

We want to warmly thank our late colleague and friend, Professor Henrik Breuning-Madsen,
who passed away during the preparation of this manuscript. He has been a key figure in moving
European soil science forward over more than three decades. This work was financially
supported by the European Union through the EC Joint Research Centre. We thank all national
stakeholders for their contributions to the development of the SPADE database. For a full list of
stakeholders we refer to ESDAC's homepage <a href="http://esdac.jrc.ec.europa.eu/">http://esdac.jrc.ec.europa.eu/</a>.

**514 **References**

[revised manuscript text omitted]

Environmental science & policy, 54, pp.438-447.

| 6 | 6 | 6 |
|---|---|---|
| - | - | _ |

| 667 | Panagos, P., Hiederer, R., Van Liedekerke, M., Bampa, F., 2013. Estimating soil organic carbon   |
|-----|--------------------------------------------------------------------------------------------------|
| 668 | in Europe based on data collected through an European network. Ecological Indicators 24, 439-    |
| 669 | 450.                                                                                             |
| 670 |                                                                                                  |
| 671 | Panagos, P., Van Liedekerke, M., Jones, A., Montanarella, L., 2012. European Soil Data Centre:   |
| 672 | Response to European policy support and public data requirements. Land Use Policy 29, 329-       |
| 673 | 338.                                                                                             |
| 674 |                                                                                                  |
| 675 | Peng, J., Loew, A. Merlin, O., Verhoest, N.E.C., 2017. A review of spatial downscaling of        |
| 676 | satellite remotely sensed soil moisture. Reviews of Geophysics 55, 341-366.                      |
| 677 |                                                                                                  |
| 678 | Platou, S.W., Nørr, A.H. & Breuning-Madsen, H., 1989. Digitisation of the EC soil map: 12 24.    |
| 679 | In: Jones, R.J.A. & Biagi, B. (ed.): Computerization of land use data. Proceedings of a          |
| 680 | symposium in the community programme for coordination of agricultural research. 20 22 May        |
| 681 | 1987, Pisa. Report EUR 11151, Luxembourg, pp 155.                                                |
| 682 |                                                                                                  |
| 683 | Pouladi, N., Møller, A. B., Tabatabai, S., & Greve, M. H. (2019). Mapping soil organic matter    |
| 684 | contents at field level with Cubist, Random Forest and kriging. Geoderma, 342, 85-92.            |
| 685 |                                                                                                  |
| 686 | Robinson, D.A., Panagos, P., Borrelli, P., Jones, A., Montanarella, L., Tye, A. & Obst C.G. Soil |
| 687 | natural capital in Europe; a framework for state and change assessment. Scientific Reports 7,    |
| 688 | 6706. DOI:10.1038/s41598-017-06                                                                  |
|     |                                                                                                  |

| 690 | Schjønning, P., van den Akker, J.J.H., Keller, T., Greve, M.H., Lamandé, M., Simojoki, A.,     |
|-----|------------------------------------------------------------------------------------------------|
| 691 | Stettler, M., Arvidsson J. & Breuning-Madsen, H., 2015. Driver-Pressure-State-Impact-Response  |
| 692 | (DPSIR) analysis and risk assessment for soil compaction – a European perspective. In: Sparks, |
| 693 | D.L. (Ed.), Advances in Agronomy, pp. 183–237. Academic Press, ISBN: 9780128030523.            |
| 694 |                                                                                                |
| 695 | SMSS/USDA/AID, 1983. Keys to Soil Taxonomy. Soil Management Support Services.                  |
| 696 | Technical Monograph no. 6. Prepared by Agronomy Department, Cornell University, Ithaca,        |
| 697 | NY, US. pp. 244.                                                                               |
| 698 |                                                                                                |
| 699 | Stockmann, U., Padarian, J. McBratney, A., Minasny, B., de Brogniez, D., Montanarella, L.,     |
| 700 | Hong, S.Y., Rawlins, B.G., Field, D.J., 2015. Global soil organic carbon assessment. Global    |
| 701 | Food Security 6, 9-16.                                                                         |
| 702 |                                                                                                |
| 703 | Sustainable Food Trust, 2015. Soil degradation: a major threat to humanity. Sustainable Food   |
| 704 | Trust, Bristol, UK.                                                                            |
| 705 |                                                                                                |
| 706 | Vossen, P., 1993. Forecasting of national crop production: The methodologies developed in the  |
| 707 | Joint Research Centre in support to the Commission of European Communities. Advance in         |
| 708 | Remote Sensing 2: 158-165.                                                                     |
| 709 |                                                                                                |

- 710 Zeraatpisheh, M., Ayoubi, S., Jafari, A., Tajik, S., & Finke, P. (2019). Digital mapping of soil
- 711 properties using multiple machine learning in a semi-arid region, central Iran. Geoderma 338,
- 712 445-452. doi: 10.1016/j.geoderma.2018.09.006
- 713
- 714 Zhang, G.-L. Liu, F., Song, X.-D., 2017. Recent progress and future prospect of digital soil
- mapping: A review. Journal of Integrative Agriculture 16, 2871-2885.
- 716
- 717

- **Table 1:** Average bulk densities calculated from the SPADE 2 database. The mean, standard
- 719 deviation and the number of observations (n) are shown.

| OM     | Depth  | Bulk Density | Std. dev.          | n   |
|--------|--------|---------------------|--------------------|-----|
| %      | cm     | g cm -3  | g cm -3 |     |
| 90-100 |        | 0.1                 | 0.13               | 165 |
| 80-90  |        | 0.1                 | 0.05               | 81  |
| 70-80  |        | 0.2                 | 0.11               | 64  |
| 60-70  |        | 0.2                 | 0.13               | 36  |
| 50-60  |        | 0.3                 | 0.13               | 25  |
| 40-50  |        | 0.4                 | 0.08               | 28  |
| 30-40  |        | 0.4                 | 0.17               | 19  |
| 20-30  |        | 0.8                 | 0.31               | 35  |
| 10-20  |        | 1.0                 | 0.72               | 176 |
| 5-10   |        | 1.1-1.2             | n/a                | n/a |
| <5     | 0-25   | 1.3                 | 0.18               | 400 |
|        | 25-50  | 1.4                 | 0.18               | 726 |
|        | 50-75  | 1.4                 | 0.17               | 719 |
|        | 75-100 | 1.5                 | 0.14               | 468 |
|        | >100   | 1.5                 | 0.18               | 714 |

Table 2: The origin of SPADE data at the national level. *Original* shows the soil profiles to
which the stakeholders originally provided data; *Profiles from other countries* show the soil
profiles for which data was copy-pasted from a similar country; *Modified profiles* show the soil
profiles to which slight adjustments were made; *Level 1 Total* shows the total number of
dominating soil profiles, which are available in the current database (SPADE-14); *Level 2 Total*(gray column) shows the total number of profiles, when associated soil types were included. The
datasets for associated soils will be available when the level 2-database (SPADE-18) is fully

730 developed.

| Country | Country         | Original  | Profiles from   | Modified   | Level 1    | Level 2    |
|---------|-----------------|-----------|-----------------|------------|------------|------------|
| code    |                 | _         | other countries | profiles   | Total      | Total      |
|         |                 | (SPADE 8) | (SPADE 14)      | (SPADE 14) | (SPADE 14) | (SPADE 18) |
| AL      | Albania         | 14        | 13              | 3          | 30         | 49         |
| AT      | Austria         | 0         | 23              | 4          | 27         | 35         |
| BE      | Belgium         | 42        | 14              | 0          | 56         | 74         |
| BG      | Bulgaria        | 0         | 16              | 7          | 23         | 40         |
| СН      | Switzerland     | 28        | 2               | 7          | 37         | 51         |
| CZ      | Czech Rep.      | 0         | 19              | 7          | 26         | 73         |
| DE      | Germany         | 60        | 15              | 2          | 77         | 149        |
| DK      | Denmark         | 13        | 0               | 0          | 13         | 29         |
| EE      | Estonia         | 11        | 2               | 4          | 17         | 26         |
| ES      | Spain           | 26        | 15              | 8          | 49         | 65         |
| FI      | Finland         | 6         | 1               | 0          | 7          | 12         |
| FR      | France          | 118       | 35              | 22         | 175        | 230        |
| GB      | United Kingdom  | 41        | 15              | 6          | 62         | 141        |
| GR      | Greece          | 10        | 15              | 4          | 29         | 66         |
| HU      | Hungary         | 40        | 10              | 11         | 61         | 92         |
| IE      | Ireland         | 18        | 4               | 3          | 25         | 44         |
| IT      | Italy           | 21        | 11              | 9          | 41         | 91         |
| LT      | Lithuania       | 0         | 20              | 8          | 28         | 52         |
| LU      | Luxembourg      | 0         | 10              | 2          | 12         | 26         |
| LV      | Latvia          | 26        | 0               | 0          | 26         | 39         |
| NL      | The Netherlands | 20        | 12              | 0          | 32         | 42         |
| NO      | Norway          | 15        | 0               | 1          | 16         | 23         |
| PL      | Poland          | 0         | 28              | 12         | 40         | 63         |
| PT      | Portugal        | 18        | 10              | 4          | 32         | 66         |
| RO      | Romania         | 28        | 28              | 21         | 77         | 115        |
| SE      | Sweden          | 0         | 9               | 3          | 12         | 23         |
| SK      | Slovakia        | 17        | 6               | 1          | 24         | 73         |
| SL      | Slovenia        | 0         | 15              | 9          | 24         | 31         |

|     | Total | 572 (31%) | 348 (19%) | 158 (9%) | 1078 (59%) | 1820 (100%) |
|-----|-------|-----------|-----------|----------|------------|-------------|
| 731 |       |           |           |          |            |             |

**Figure 1**: a) Structure of the European Soil Database to which SPADE provides data (after

Lambert et al., 2003), b) Timeline of the establishment of the Soil Profile Analytical Database of

736 Europe (SPADE). See text for details.

---

## Author Response (AR1)

**Response to the topical editor 2019-08-20**

Dear editor.

Thanks for your kind reception of our manuscript and the proposed modifications following reviewer comments. We hope you will find it fit for publications after the following revisions.

We have attached a separate pdf version of Fig1, as we found it hard to read in the auto-generated version. Do not hesitate to contact us if you need further information or clarification or need higher resolution versions of the figures.

Here are our responses to your comments (in red):

Comment 1:

Comment 7: I coyuld not find the correction in the text.

Response: the line reference in our response to reviewer 1 was to an intermediate version of the manuscript, which was submitted as a supplementary to the responses to reviewer 1, hence it did not correspond to the final version of the manuscript. We were not fully confident with the alternative open review process, so we may have sent too many intermediate versions. We apologise for any inconvenience, and hope you will find the relevant text in line 225 onwards in the attached final and updated version of the manuscript. Otherwise, we refer to the full report sent to the stakeholders (Breuning-Madsen et al. 2015).

Comment 2:

Comment 14 Correcting for coarse fragments has recently been discussed in SOIL by Poeplau et al and Hobley et al. Please note that the gravimetric coarse particle fraction needs to be used. You did not specify 'gravimetric'.

Response: We used the gravimetric coarse soil fraction, which has now been specified in the text (line 380-381). Further, we have added a reference to Hobley et al. 2018.